# Profilin and formin constitute a pacemaker system for robust actin filament growth

Johanna Funk[1], Felipe Merino[2], Larisa Venkova[3], Lina Heydenreich[4], Jan Kierfeld[4], Pablo Vargas[3], Stefan Raunser[2], Matthieu Piel[3], Peter Bieling[1]*

[1]Department of Systemic Cell Biology, Max Planck Institute of Molecular Physiology, Dortmund, Germany; [2]Department of Structural Biochemistry, Max Planck Institute of Molecular Physiology, Dortmund, Germany; [3]Institut Curie UMR144 CNRS, Paris, France; [4]Physics Department, TU Dortmund University, Dortmund, Germany

**Abstract** The actin cytoskeleton drives many essential biological processes, from cell morphogenesis to motility. Assembly of functional actin networks requires control over the speed at which actin filaments grow. How this can be achieved at the high and variable levels of soluble actin subunits found in cells is unclear. Here we reconstitute assembly of mammalian, non-muscle actin filaments from physiological concentrations of profilin-actin. We discover that under these conditions, filament growth is limited by profilin dissociating from the filament end and the speed of elongation becomes insensitive to the concentration of soluble subunits. Profilin release can be directly promoted by formin actin polymerases even at saturating profilin-actin concentrations. We demonstrate that mammalian cells indeed operate at the limit to actin filament growth imposed by profilin and formins. Our results reveal how synergy between profilin and formins generates robust filament growth rates that are resilient to changes in the soluble subunit concentration.

DOI: https://doi.org/10.7554/eLife.50963.001

*For correspondence:
peter.bieling@mpi-dortmund.mpg.de

Competing interests: The authors declare that no competing interests exist.

## Introduction

Eukaryotic cells move, change their shape and organize their interior through dynamic actin networks. Actin assembly requires nucleation of filaments, which elongate by the addition of subunits to filament ends. To move and quickly adapt their shape, most eukaryotic cells sustain vast amounts (>50 μM) of polymerizable subunits, which requires the monomer-binding protein profilin (*Koestler et al., 2009*; *Pollard et al., 2000*; *Raz-Ben Aroush et al., 2017*; *Skruber et al., 2018*). Profilin shields the barbed end side of actin monomers to suppress spontaneous nucleation (*Schutt et al., 1993*). This allows profilin-actin complexes to exist at high concentrations in vivo, unlike free actin monomers. Profilin-actin is therefore considered the physiological substrate of filament growth (*Kaiser et al., 1999*; *Pantaloni and Carlier, 1993*; *Pollard et al., 2000*) which occurs when profilin-actin complexes bind to exposed filament barbed ends (*Gutsche-Perelroizen et al., 1999*; *Kinosian et al., 2002*; *Pollard and Cooper, 1984*; *Pring et al., 1992*). The speed of filament elongation over a limited concentration range of profilin-actin fits a linear model for a binding-controlled reaction (*Blanchoin and Pollard, 2002*; *Oosawa and Asakura, 1975*). This has led to the idea that the concentration of soluble subunits is the central parameter that controls the speed of actin growth (*Blanchoin et al., 2014*; *Carlier and Shekhar, 2017*; *Pollard et al., 2000*). However, actin elongation has only been studied at low, non-physiological levels of soluble subunits until now.

The concentration of profilin-actin is thought to pace not only spontaneous, but also catalyze actin growth by actin polymerases such as formins (*Paul and Pollard, 2009*). These modular proteins

bind the filament barbed end via their FH2 domain and recruit many profilin-actin complexes through flexible FH1 domains (*Figure 1A*). Polymerase activity is thought to arise from formins ability to increase the binding frequency of profilin-actin to growing filament ends (*Courtemanche, 2018*; *Paul and Pollard, 2009*; *Vavylonis et al., 2006*). Whether, however, filament growth in vivo is controlled at the level of binding is unknown. Consequently, we do not fully understand how formins function as actin polymerases in cells.

The model of linear concentration-dependent scaling of actin growth creates a conundrum because of two reasons: i) Filament growth from profilin-actin complexes cannot occur in a single binding step, but requires additional reactions whose rate should not depend on the free subunit concentration (*Figure 1A*). Binding of profilin-actin to the actin filament barbed end occludes the binding site for new subunits and profilin needs to be released for elongation to continue (*Figure 1B*) (*Courtemanche and Pollard, 2013*; *Pernier et al., 2016*; *Pollard and Cooper, 1984*). How rapidly profilin release occurs and whether it affects filament growth is presently unclear (*Blanchoin and Pollard, 2002*; *Gutsche-Perelroizen et al., 1999*; *Romero et al., 2004*). ii) Generally, soluble actin concentrations vary significantly across species, cell types (*Koestler et al., 2009*; *Pollard et al., 2000*; *Raz-Ben Aroush et al., 2017*) and likely even within a single cell (*Skruber et al., 2018*). If elongation rates scale linearly with profilin-actin concentrations, then actin filaments must grow at widely different speeds in vivo. Actin polymerases like formins should dramatically amplify such variations. This poses a conceptual challenge to the construction of functional actin networks whose architecture should directly depend on the filament elongation speed. We presently do not understand whether or how cells control the rate of filament growth when facing variable and fluctuating profilin-actin levels. Here we uncover a mechanism that establishes intrinsically robust, but tunable growth rates that are buffered against changes in the free subunit concentration.

## Results

### Actin filament growth at physiological profilin-actin concentrations

To reconstitute actin assembly at cell-like conditions, we first determined the concentration of actin and the two most abundant profilin isoforms (−1 and −2) (*Mouneimne et al., 2012*) in mammalian cells through volume measurements (*Cadart et al., 2017*) and western blots (*Figure 1C–D*, *Figure 1—figure supplement 1A*, *Figure 1—figure supplement 2*, Materials and methods). We studied mesenchymal (HT1080), epithelial (B16F10) or immune cells (T-cells, dendritic cells and neutrophils), the latter because of their rapid motility. Consistent with earlier estimates (*Pollard et al., 2000*; *Witke et al., 2001*) profilin and actin were highly expressed (*Figure 1D*, *Figure 1—figure supplement 1A*). Profilin-1 was the dominant isoform, whereas profilin-2 was not present at substantial levels in most cells (*Figure 1—figure supplement 1A*). Profilin levels were especially high in immune cells, in line with their overall fast motility (*Lämmermann and Sixt, 2009*; *Vargas et al., 2017*) and their ability to very rapidly assemble actin-rich pseudopods especially in low-adhesive environments (*Lämmermann et al., 2008*; *Renkawitz et al., 2009*). Actin always exceeded the profilin concentration as expected, since actin forms filaments and binds monomer-binding proteins other than profilin. Because profilin binds mammalian cytoplasmic actin much more tightly than other abundant monomer-binding proteins like thymosin-$\beta_4$ (see below), the actin pool is likely sufficiently large for profilin to be nearly completely bound to monomers in vivo (*Kaiser et al., 1999*). We thus estimated the profilin-actin concentration around 50–200 µM, depending on mammalian cell type (*Figure 1D–E*, see Materials and methods section for details concerning the estimation of soluble profilin-actin levels).

To study actin elongation at these conditions, we first adapted methods (*Hatano et al., 2018*; *Ohki et al., 2009*) to purify mammalian cytoplasmic actin (β-γ isoforms). Past studies relied mostly on muscle α-actin, the most divergent actin isoform. Its widespread use, combined with chemical labeling, created confusion concerning the role of profilin in the past (*Blanchoin and Pollard, 2002*; *Courtemanche and Pollard, 2013*; *Kinosian et al., 2002*; *Kinosian et al., 2000*; *Pernier et al., 2016*; *Romero et al., 2007*; *Vavylonis et al., 2006*). To study the authentic substrate of actin assembly in non-muscle cells, we purified either i) native bovine actin from thymus tissue or ii) recombinant human β-actin from insect cells (*Figure 1—figure supplement 1B*, Materials and methods). Using

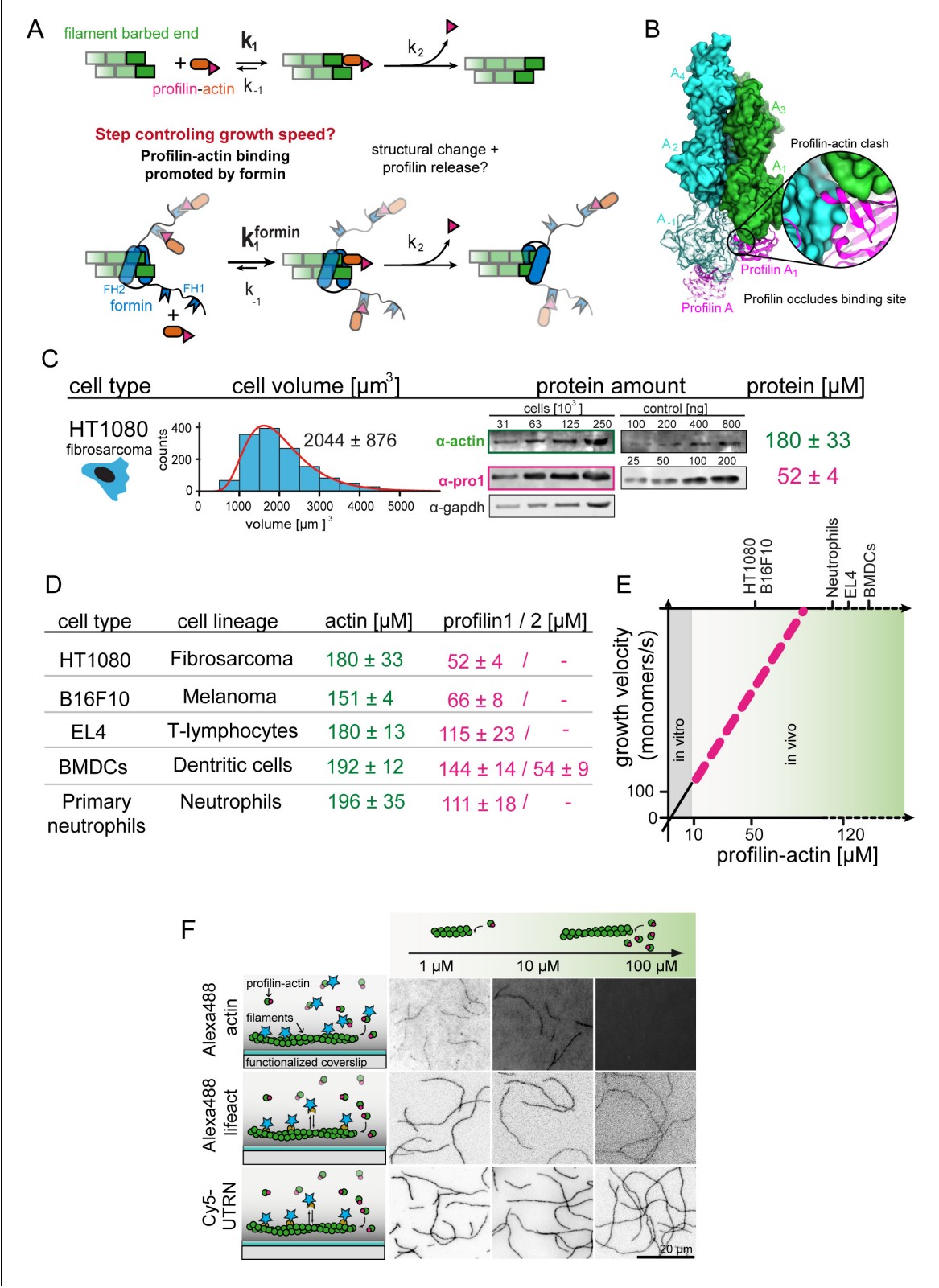

**Figure 1.** Filament assembly at physiological profilin-actin concentrations. (**A**) Scheme of barbed end elongation from profilin-actin alone (top) or with formins (bottom). (**B**) Structural model of profilin at filament barbed ends (Materials and methods). The incoming profilin-actin complex is transparent. Actin is shown as green surface while profilin as magenta ribbons. Inset highlights the clash between the incoming actin monomer and profilin. (**C**) Profilin-actin measurements in HT1080 cells. Left to right: single cell volume histogram, western blots of actin, profilin1 (left: cell titration, right: standard

*Figure 1 continued on next page*

*Figure 1 continued*

curve of recombinant proteins), values are mean (N = 3) and SD, Materials and methods. (D) Table of total concentrations of actin and profilin-1/2 in various mammalian cell types (*Figure 1—figure supplement 1*). (E) Scheme of a linearly substrate-dependent actin elongation rate. Top axis: Profilin-actin amounts for various cell types as indicated. (F) Scheme (left) and TIRFM images (right) of elongating filaments at indicated profilin-actin concentrations visualized with top- Alexa488-labeled monomers (20% labeled), middle - 10 nM Alexa488-lifeact, bottom – 10 nM Cy5-UTRN$_{261}$.

DOI: https://doi.org/10.7554/eLife.50963.002

The following source data and figure supplements are available for figure 1:

**Source data 1.** Data *Figure 1*.
DOI: https://doi.org/10.7554/eLife.50963.005
**Figure supplement 1.** Quantification of profilin-actin levels and purification of mammalian profilin-actin.
DOI: https://doi.org/10.7554/eLife.50963.003
**Figure supplement 2.** Western blots and graphical summary of profilin-actin levels per cell.
DOI: https://doi.org/10.7554/eLife.50963.004

mass spectrometry, we detected β and γ actin in a roughly 1:1 ratio, but no α-actin in native actin. Mammalian cytoplasmic actin in the absence of profilin polymerized at speed that linearly depended on the monomer concentration up to 10 μM as expected (*Figure 1—figure supplement 1C*). Association rates determined from linear fits to this data were comparable to actin from other organisms (*Bieling et al., 2018*; *Pollard, 1986*). Higher monomeric actin concentrations could not be explored due to the well-known propensity of bare actin monomers to spontaneously nucleate filaments.

We then studied binding of the most abundant monomer-binding proteins, profilin-1/–2 and thymosin-β$_4$, to ATP-bound mammalian cytoplasmic actin (*Figure 1—figure supplement 1D–E*). In general agreement with studies using non-muscle actin (*Bieling et al., 2018*; *Kinosian et al., 2002*; *Vinson et al., 1998*), thymosin-β$_4$ bound weakly ($K_D$ ~1.2 μM), whereas profilin bound exceptionally strongly ($K_D$ ~18 nM) to ATP-bound actin monomers at near-physiological ionic strength (0.133 M, see Materials and methods). This allowed us to isolate heterodimeric complexes of profilin and ATP-bound actin by size-exclusion chromatography (*Figure 1—figure supplement 1F*) and to concentrate them (>500 μM) without triggering nucleation. We then turned to total internal-reflection fluorescence microscopy (TIRFM) assays to analyze elongation of surface-tethered actin filaments (*Figure 1F*, *Figure 2A*). Strong background prevented us from using fluorescent actin at high concentrations (*Figure 1F* upper). Trace amounts (10 nM) of fluorescent filament-binding probes (UTRN$_{261}$ or LifeAct), however, yielded sufficient contrast without altering assembly kinetics (*Figure 1F* middle and lower, [*Bieling et al., 2018*]). To further minimize nucleation, we additionally added low amounts of either free profilin (<2 μM) or thymosin-β$_4$ (<15 μM) at high profilin-actin concentrations (Materials and methods). As expected, this did not impact filament elongation (*Figure 2—figure supplement 1A–B*). These advances allowed us to, for the first time, study mammalian cytoplasmic actin growth over the entire physiological range of profilin-actin concentrations.

## Profilin dissociation kinetically limits filament elongation

As previously (*Blanchoin and Pollard, 2002*; *Jégou et al., 2013*), we observed a linear increase of the actin filament growth velocity at low profilin-actin concentrations (<10 μM, *Figure 2B–C*,

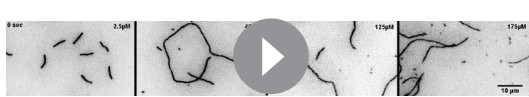

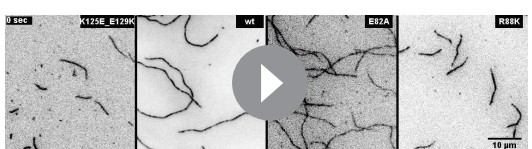

**Video 1.** Polymerization of actin filaments from different profilin1-actin concentrations. Filaments were visualized with 10 nM Cy5-UTRN$_{261}$ in TIRF-M. Polymerization from increasing profilin1-actin concentrations from left to right: 2.5 μM, 40 μM, 125 μM, 175 μM.
DOI: https://doi.org/10.7554/eLife.50963.009

**Video 2.** Polymerization of actin filaments from different profilin1 mutant-actin complexes at 125 μM. Filaments were visualized with 10 nM Cy5-UTRN$_{261}$ in TIRF-M. Polymerization was performed from the following profilin1 mutant-actin complexes, left to right: profilin1-K125E + E129K, -wt, -E82A, -R88K.
DOI: https://doi.org/10.7554/eLife.50963.013

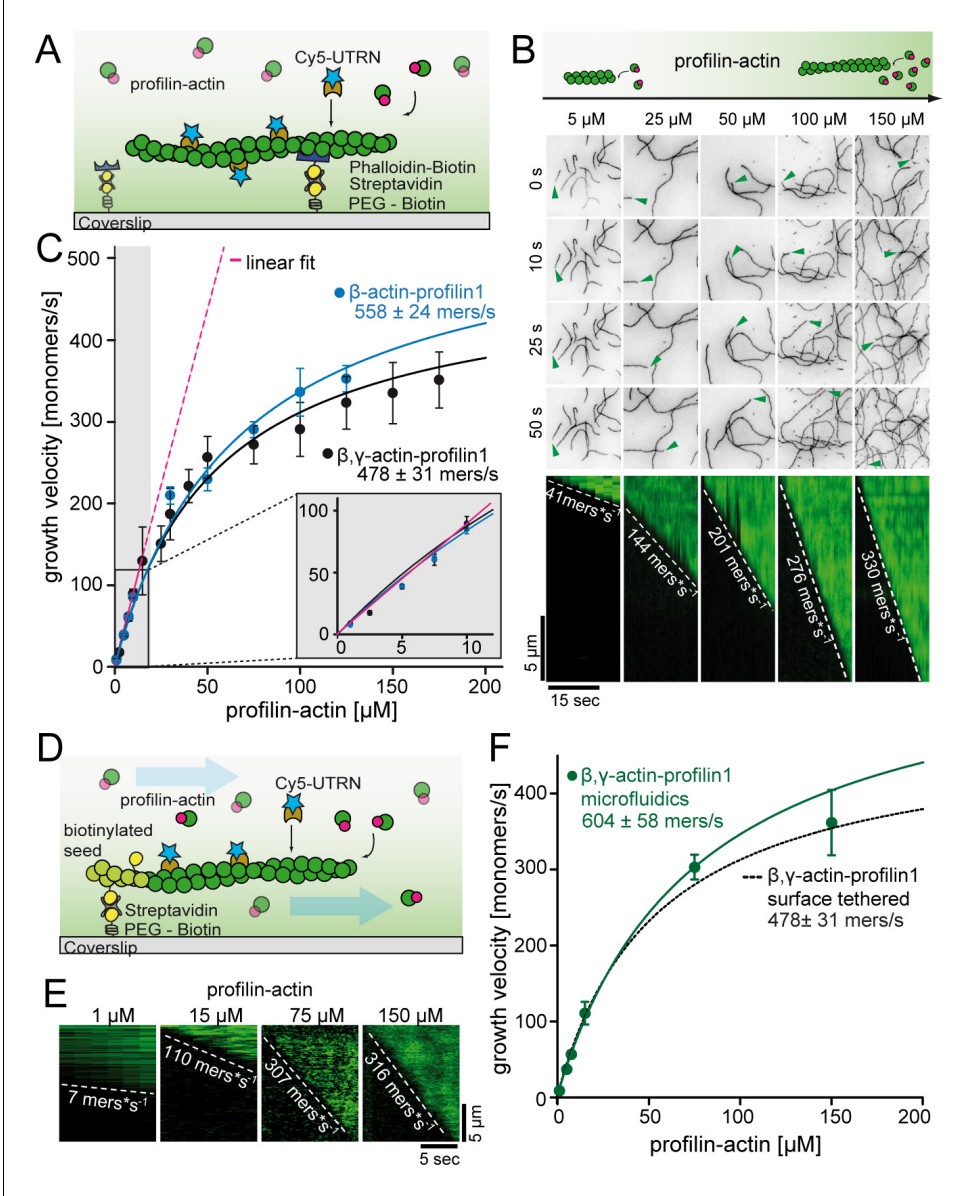

**Figure 2.** A kinetic limit to actin filament elongation from profilin-actin. (**A**) Scheme of TIRFM elongation assays of surface-attached filaments from profilin-actin on functionalized coverslips. (**B**) TIRFM time-lapse images (top) and kymographs (bottom) of filament elongation (green arrow follows a single barbed end) at indicated profilin-actin concentrations. (**C**) Barbed end growth velocities from TIRFM assays using different profilin1:actin complexes as indicated. Points are mean values [N ≥ 40 for each condition, error = SD]. Lines are hyperbolic fits. Inset: Regime of low concentrations fitted by a linear model (magenta, *Figure 2—figure supplement 1C–D*). (**D**) Scheme of microfluidic experiments of seed-attached filaments under flow. (**E**) Kymographs of filaments at indicated profilin-actin concentrations in microfluidic experiments. (**F**) Barbed end growth velocities of filaments grown in microfluidic channels in TIRFM assays (green) compared to surface tethered filaments as quantified in ((**C**), black dashed line). Points are mean values [N ≥ 40 for each condition, error = SD].

DOI: https://doi.org/10.7554/eLife.50963.006

The following source data and figure supplement are available for figure 2:

**Source data 1.** Data *Figure 2*.
DOI: https://doi.org/10.7554/eLife.50963.008

**Figure supplement 1.** Control experiments for barbed end polymerization in TIRF-M single filament assays.
DOI: https://doi.org/10.7554/eLife.50963.007

*Figure 2—figure supplement 1C–D*). Strikingly, however, elongation rates deviated strongly from linearity at moderate (>20 μM) and nearly saturated at high (≥100 μM) concentrations to plateau at ~500 monomers s$^{-1}$ (*Figure 2B–C*, *Video 1*). Data could be well fitted with Michaelis Menten kinetics assuming a binding reaction followed by a rate-limiting step. Importantly, the maximal elongation rate did not depend on surface tethering, the filament-binding probe or the specific cytoplasmic isoform of profilin or actin (*Figure 2—figure supplement 1C–E*). We ruled out accumulation of free profilin as a reason for saturation, because filament growth was constant over time under all conditions (*Figure 2—figure supplement 1A–B*). We observed saturation also in microfluidic assays with a constant influx of fresh profilin-actin for filaments that were only attached via short seeds (*Figure 2D–F*, *Figure 2—figure supplement 1*). This demonstrates that filament elongation at physiological conditions is not controlled by the diffusion-limited association of profilin-actin to barbed ends, but is kinetically limited by a reaction that proceeds with a rate of ~500 s$^{-1}$.

Structural models suggest that incorporation of profilin-actin transiently caps barbed ends, because profilin sterically hinders the binding of the next monomer (*Figure 1B*, [*Courtemanche and Pollard, 2013*]). Profilin release is therefore required for continual elongation (*Figure 3A*). Profilin binds much more weakly to filament barbed ends than to monomeric actin (*Courtemanche and Pollard, 2013*; *Pernier et al., 2016*). We confirmed that profilin dissociation from actin monomers ($k_{off}$ = 0.77 s$^{-1}$, *Figure 3—figure supplement 1B–D*) is much slower than the maximal elongation rates we observe (~500 s$^{-1}$). This means that structural changes in the terminal actin protomer are required to trigger profilin release. We deduced that either of these subsequent reactions could become rate-limiting (*Figure 3A*). Profilin dissociation specifically, should be affected by interactions between actin and profilin. To test this hypothesis, we introduced mutations in profilin-1 at the actin binding interface to either decrease (E82A, R88K) or increase (K125E+E129K) affinity (*Figure 3B–C, F*, *Figure 3—figure supplement 1A*, Materials and methods). Single point mutants (E82A and R88K) caused a moderate reduction (~1.5 and ~4– fold, respectively), while mutation of two residues (K125E+E129K) showed an increase (~5–fold) in monomer binding affinity (*Figure 3C*). Importantly, these changes were caused by altered monomer dissociation, but not association rate constants (*Figure 3F*, *Figure 3—figure supplement 1B–D*). More drastic changes were incompatible with elongation assays due to either accumulation of free actin (severely weakening mutants) or the complete inhibition of growth (ultra-tight binding mutants, *Figure 3—figure supplement 1E–F*).

We then tested the effect of these profilin mutations on filament growth. Strikingly, the maximal elongation rate scaled with the monomer dissociation rate of profilin. Weakly-binding profilins increased, whereas tight-binding profilin reduced the maximal filament growth rate (*Figure 3D–F*, *Video 2*). To better understand the effects of these profilin mutations on the elongation reaction, we developed an analytical model of actin polymerization in the presence and absence of profilin (see Appendix Section). Fitting this model to the data revealed that our profilin mutations altered both the binding rate of profilin-actin to the barbed end as well as the rate of release of profilin following polymerization. The latter determines the rate of filament elongation at saturation. We draw two conclusions from these observations: i) The profilin mutations impact the dissociation of profilin from both soluble actin monomers and terminal actin subunits similarly. ii) The strength of the profilin-actin interaction modulates the rate-limiting step of elongation. This strongly suggests that profilin dissociation from the barbed end imposes a kinetic limit to actin filament elongation.

Some previous studies have linked profilin release from barbed ends to the rapid hydrolysis of ATP within actin (*Pernier et al., 2016*; *Romero et al., 2004*). We therefore generated ATPase-deficient (AD) actin, by mutations of three residues within the catalytic core of actin (Q137A+D154A+H161A, *Figure 4A*). These combined mutations did not abolish nucleotide binding, affect polymerization or reduce the affinity for profilin (*Figure 4B–E*, *Figure 4—figure supplement 1A*). Endpoint (*Figure 4B*) and time-resolved ATPase assays (*Figure 4C*) showed that this triple mutant was indeed unable to hydrolyze its associated ATP nucleotide with appreciable rates even after polymerization from profilin-actin and therefore formed filaments that were exclusively and homogenously ATP bound. Importantly, we found that ATPase-deficient actin was able to elongate actin filaments with nearly the same rates as wildtype actin at saturating profilin-actin concentrations (*Figure 4D–E*). This clearly demonstrates that profilin release from the barbed end does not require cleavage of the β-γ phosphodiester bond of ATP in actin. More generally, the lack of assembly-related defects for ATPase deficient actin is consistent with the notion that ATP hydrolysis serves an essential function unrelated to filament assembly (*De La Cruz et al., 2000*; *Pollard et al., 2000*).

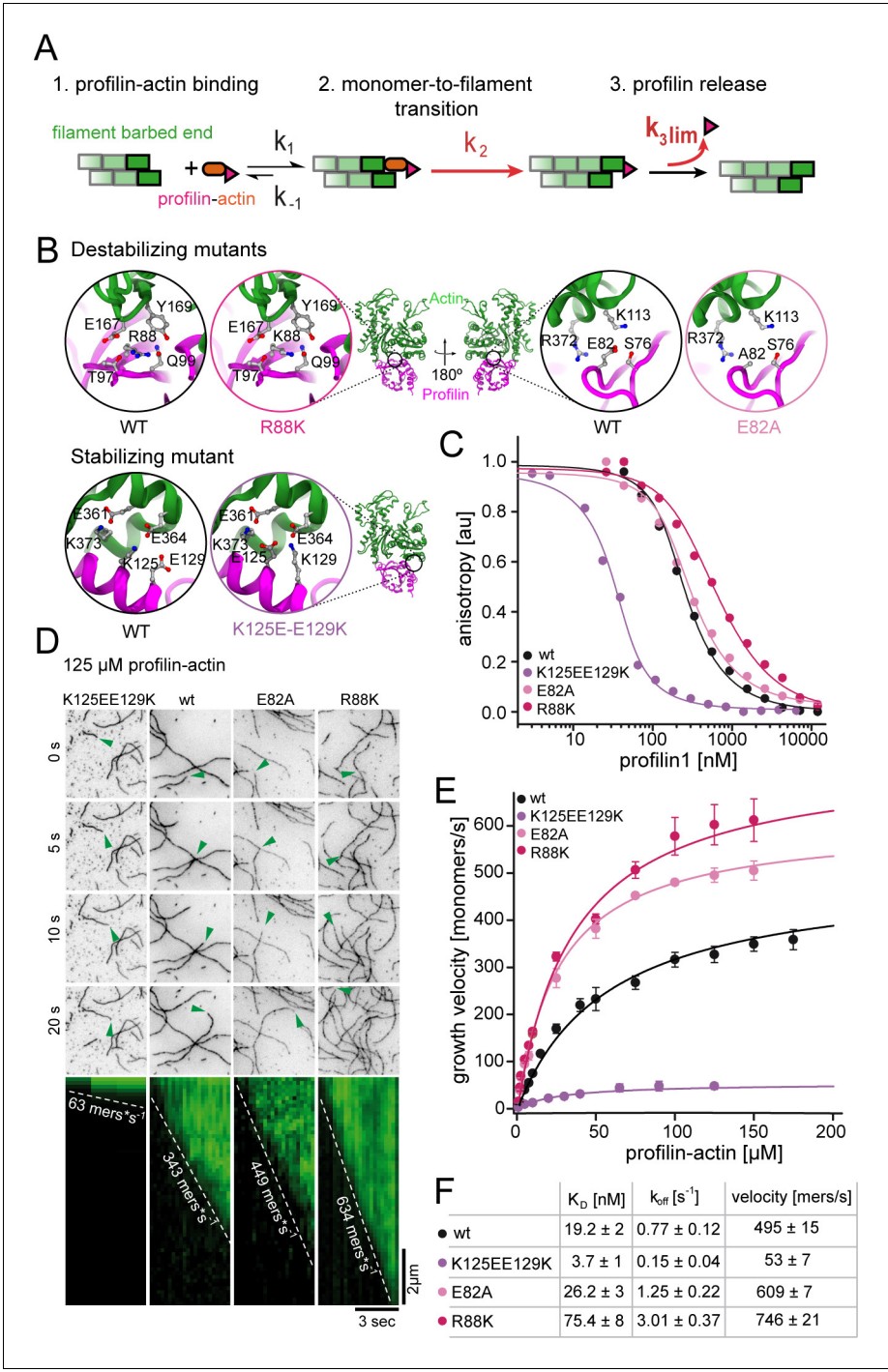

**Figure 3.** Profilin release kinetically limits filament elongation. (**A**) Scheme of barbed end elongation from profilin-actin alone indicating the potential limiting kinetic steps. (**B**) Structural models (Materials and methods) of the actin interface of stabilizing and destabilizing profilin mutants. Ribbon diagrams highlight the mutation positions. Insets show changes in amino acid environments upon mutation. (**C**) Binding of profilin to ATP-bound actin monomers measured by fluorescence anisotropy competition assays. Fluorescence anisotropy of Atto488-WAVE1$_{WCA}$ (4 nM) at increasing profilin1 (wt or mutants as indicated) concentrations in the presence of actin monomers (150 nM for wt and weakly binding profilin and 40 nM for tightly binding profilin). Lines fit to an analytical competition model (Materials and methods). Points represents means (N $\geq$ 3) $\pm$ SD. (**D**) TIRFM time-lapse images (top) and kymographs (bottom) of filament elongation (green arrow follows a single barbed end) from mutant profilin1:actin complexes (125 µM total) as indicated. (**E**) Barbed end growth velocities measured from TIRFM assays using mutant profilin1:actin complexes as indicated. Points are mean values [N $\geq$ 40 for each concentration, error = SD].
*Figure 3 continued on next page*

eLIFE Research article                                    Biochemistry and Chemical Biology | Cell Biology

*Figure 3 continued*

Lines are hyperbolic fits. (**F**) Summary table of equilibrium dissociation constants ($K_D$) and dissociation rate constants ($k_{off}$, *Figure 3—figure supplement 1*) of the interaction of profilin1 (wt or mutants as indicated) and actin monomers and the resulting maximal filament elongation velocities as measured by TIRFM.
DOI: https://doi.org/10.7554/eLife.50963.010

The following source data and figure supplement are available for figure 3:

**Source data 1.** Data *Figure 3*.
DOI: https://doi.org/10.7554/eLife.50963.012

**Figure supplement 1.** Measurements of profilin1-actin monomer association kinetics and characterization of a ultra-tight binding profilin1 mutant that blocks filament polymerization.
DOI: https://doi.org/10.7554/eLife.50963.011

## Formin actin polymerases promote profilin release through their FH2 domain

Actin elongation in cells can be facilitated by actin polymerases such as formins. These proteins are thought to increase the rate of binding between profilin-actin complexes and the barbed end they processively associate with (*Paul and Pollard, 2009*). Because such a mechanism can only accelerate growth when binding is limiting, we asked how formins affect actin assembly at saturating profilin-actin concentrations. We focused on Diaphanous-type formins because of their established polymerase function. We introduced constitutively active mDia1, containing profilin-actin-interacting FH1 and barbed end-binding FH2 domains, to TIRFM assays (*Figure 5A*). We used formin concentrations sufficient to saturate filament barbed ends, as evident from their accelerated growth rate compared to control experiments (Materials and methods). We verified that the measured velocities match the speed of formins observed at the single-molecule level (*Video 3*). mDia1 strongly accelerated barbed-end growth at limiting profilin-actin concentrations ($\leq$10 µM), as expected (*Jégou et al., 2013*; *Kovar et al., 2006*). Importantly, mDia1-mediated elongation still exhibited saturation at elevated profilin-actin levels, but converged to a much higher (4x-fold) maximal rate than observed for free ends (*Figure 5B–C*, *Figure 5—figure supplement 1A*). This demonstrates that formins can accelerate the rate-limiting reaction in filament elongation at saturating profilin-actin concentrations.

To test whether this ability is shared among formins, we studied other diaphanous- (mDia2) and non-diaphanous (DAAM1) formins. Indeed, both mDia2 and DAAM1 accelerated filament elongation not only at limiting, but also saturating profilin-actin concentrations albeit less strongly than mDia1 (*Figure 5B–C*, *Video 4*). The relative rate enhancement of all formins decreased only slightly with substrate concentrations (*Figure 5—figure supplement 1A*). Formins thus slightly broaden the regime over which actin growth is insensitive to the profilin-actin concentration (*Figure 5—figure supplement 1B*).

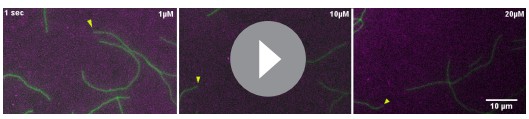

**Video 3.** Polymerization of actin filaments from profilin1-actin at different concentrations in presence of mDia1 FH1-FH2. Filaments were acquired in TIRF-M (filaments with 10 nM Cy5-UTRN$_{261}$ - green; 0.7 nM TMR-mDia1 FH-FH2 – magenta). mDia1-mediated actin filament barbed end polymerization was performed at different profilin1-actin concentrations, left to right: 1 µM, 10 µM, 20 µM. For guidance, an example of a visible labeled mDia1 molecule processively moving with a filament barbed end is highlighted with an error.
DOI: https://doi.org/10.7554/eLife.50963.020

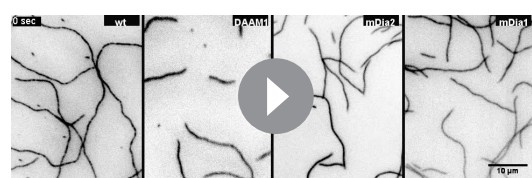

**Video 4.** Polymerization of actin filaments from 75 µM profilin1-actin in presence/absence of formins. Filaments were visualized with 10 nM Cy5-UTRN$_{261}$ in TIRF-M. All filament barbed ends were saturated with 15 nM formin FH1-FH2. Polymerization was performed in presence of different formins, left to right: wt (no formin), + DAAM1, +mDia2, +mDia1.
DOI: https://doi.org/10.7554/eLife.50963.021

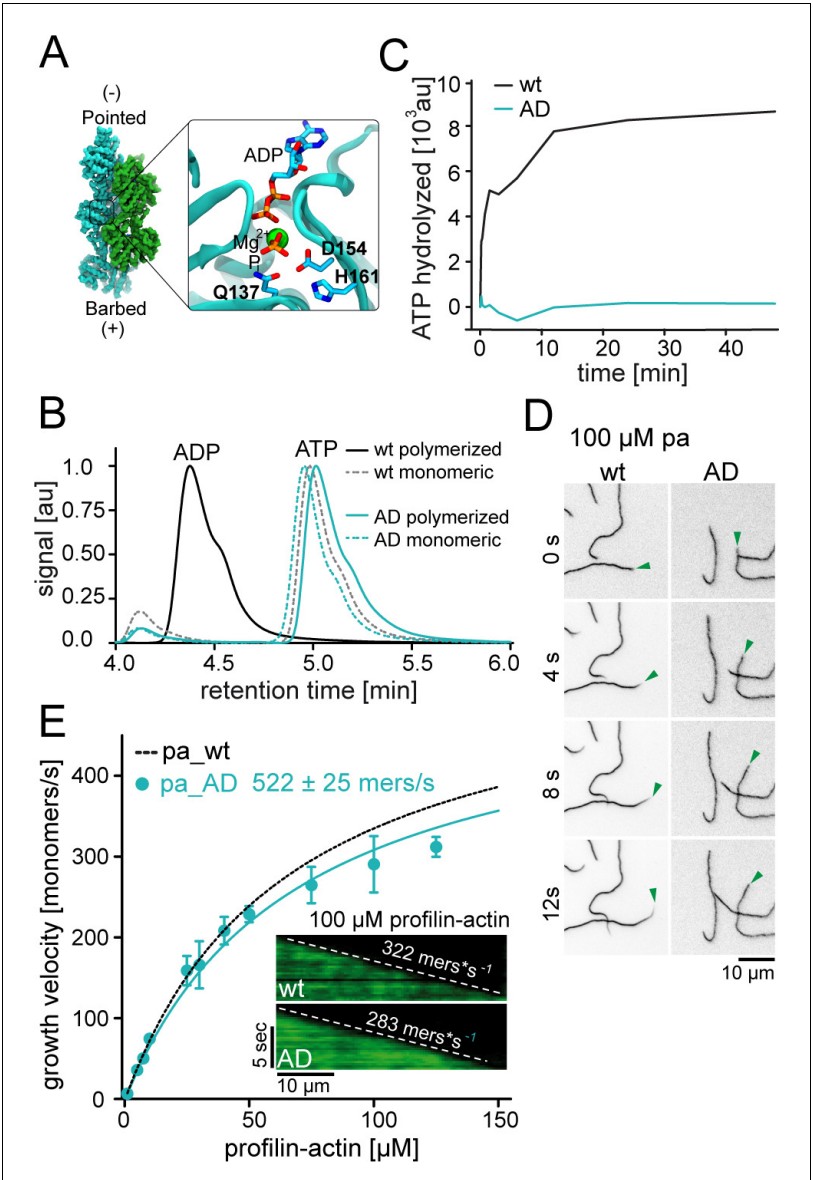

**Figure 4.** ATP hydrolysis is not required for profilin release from the barbed end. (**A**) Nucleotide-binding site of filamentous actin. Left: the overall structure of filamentous actin. Right: Inset of the active site (PDBID 6FHL), including the three amino acids involved in nucleotide hydrolysis which were mutated to alanine for the generation of ATPase deficient actin, and the products of the reaction ADP and Pi. (**B**) End-point assays examining nucleotide content via HPLC after 1.5 hr of seeded polymerization from profilin-actin (either wt or ATPase deficient). As a non-polymerized control, profilin-actin was stabilized via LatrunculinB before the experiment. (**C**) ATPase activity of wt and ATPase deficient actin in seeded polymerization assays. The cleavage of $\gamma$-$^{32}$P is monitored over time after mixing profilin1: actin complexes containing radioactive ATP with filaments in a 1:1 ratio (12 µM total)). (**D**) TIRF-M time-lapse images of filament barbed end elongation (green arrow follows a single barbed end) from either wt- or ATPase deficient actin-containing profilin1-actin complexes (100 µM total). (**E**) Barbed end growth velocities of profilin1–actin (100 µM total, wt (black) or ATPase deficient (cyan)) from TIRFM assays. Points are mean values [N $\geq$ 40 for each concentration, error = SD]. Lines are hyperbolic fits. Inset: Kymographs of filament growth.

DOI: https://doi.org/10.7554/eLife.50963.014

The following source data and figure supplement are available for figure 4:

**Source data 1.** Data *Figure 4*.

DOI: https://doi.org/10.7554/eLife.50963.016

**Figure supplement 1.** Affinity measurements of profilin1 to wt β-actin and ATPase-deficient actin.

DOI: https://doi.org/10.7554/eLife.50963.015

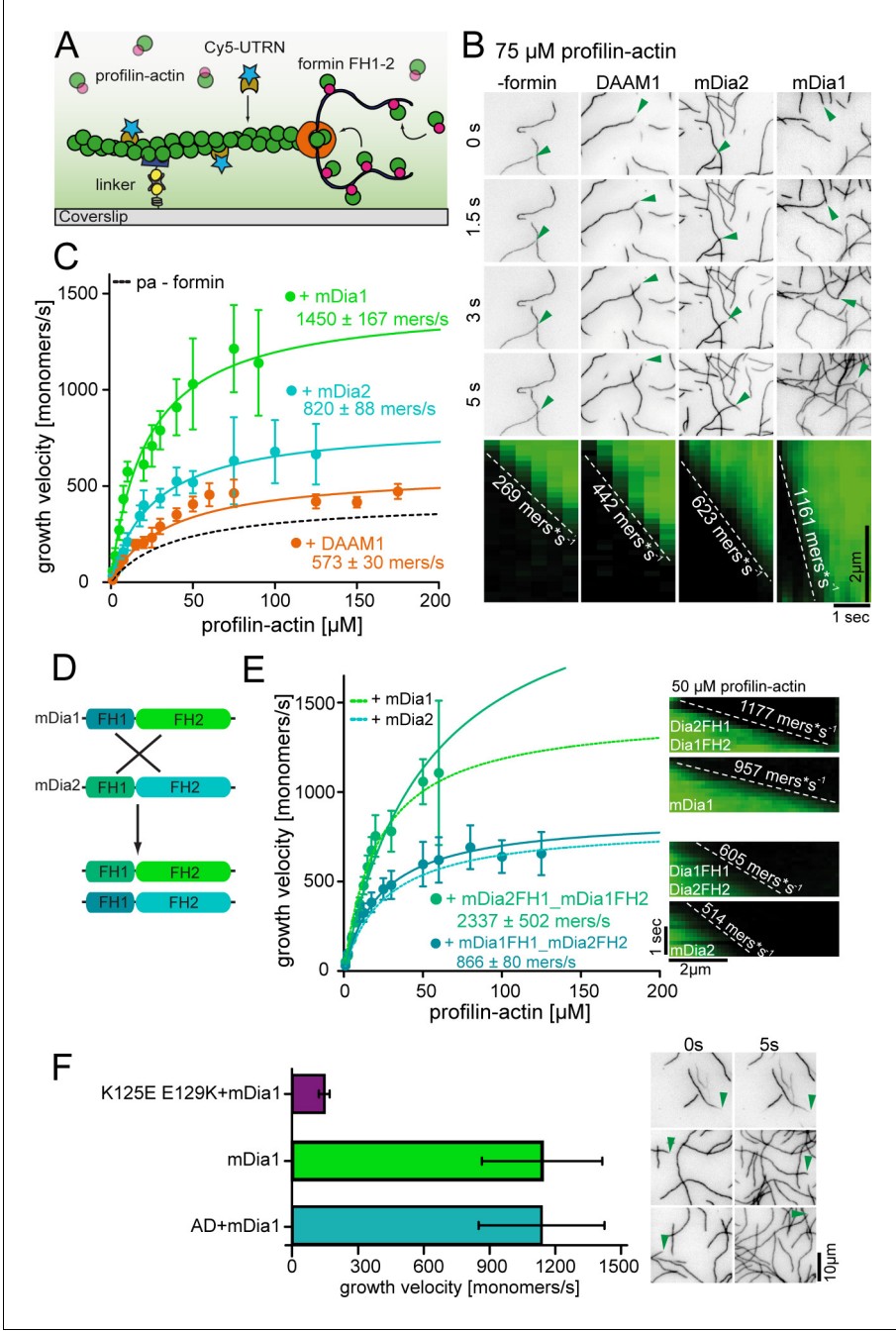

**Figure 5.** Formins accelerate filament elongation at saturating profilin-actin concentrations. (**A**) Scheme of TIRFM assays with formin catalyzing the elongation of a filament from profilin-actin on functionalized coverslips. (**B**) Top: TIRFM time-lapse images of formin-mediated actin elongation (green arrows follow a single barbed end) at 75 µM profilin-actin in the absence or in the presence of 15 nM formin constructs as indicated. Bottom: Kymographs of individual growing filaments as in the top panel. (**C**) Velocities of formin-catalyzed barbed end growth from TIRFM assays as in (**B**). Points are mean values [N ≥ 40 for each concentration, error = SD]. Lines are hyperbolic fits. (**D**) Scheme of the generation of mDia chimeras. (Materials and methods). (**E**) Barbed end growth velocities of mDia chimeras (continuous lines) compared to wt mDia formins ((**B**), dashed lines) from TIRFM assays. Points are mean values [N ≥ 40 for each condition, error = SD]. Lines are hyperbolic fits. Right: Kymographs of growing filaments (±formins as indicated) at 50 µM profilin-actin. (**F**) Comparison of mDia1(15 nM)-mediated filament growth from 100 µM profilin-actin (either both wt proteins, tight binding profilin-1 (K125E-E129K) or ATPase-deficient actin (AD) as indicated). Left: Growth velocities. Points are mean values [N ≥ 35 for each condition, error = SD]. Right: TIRFM time-lapse images (green arrows follow a single barbed end).

*Figure 5 continued on next page*

*Figure 5 continued*

DOI: https://doi.org/10.7554/eLife.50963.017

The following source data and figure supplement are available for figure 5:

**Source data 1.** Data *Figure 5*.
DOI: https://doi.org/10.7554/eLife.50963.019
**Figure supplement 1.** Profilin release but not ATP hydrolysis is limiting for formin-mediated actin polymerization.
DOI: https://doi.org/10.7554/eLife.50963.018

Interestingly, even closely related formins such as mDia1 and 2 differ in their ability to accelerate the rate-limiting reaction of filament elongation. To understand the origin of this difference, we created chimeras of mDia1 and 2 by swapping their FH1 and FH2 domains (*Figure 5D*). Both chimeras accelerated filament growth, but generated distinct maximal rates at saturating profilin-actin concentration (*Figure 5E*). Interestingly, mDia2FH1-mDia1FH2 exhibited similar maximal rates as mDia1, whereas mDia1FH1-mDia2FH2 was comparable to mDia2 (*Figure 5E*). This demonstrates that the barbed-end associated FH2 domain is responsible for setting the maximal rate of filament elongation.

Finally, to test which constraints limit formin-mediated growth, we elongated mDia1-associated actin filaments using profilin-actin complexes containing either ATPase-deficient actin or tight-binding profilin. ATPase-deficient actin grew with rates indistinguishable from wildtype actin, whereas tight profilin binding inhibited mDia1-mediated growth (*Figure 5F*, *Figure 5—figure supplement 1C*). This demonstrates that formin-mediated filament elongation at saturation is limited by profilin release from the barbed end and not nucleotide hydrolysis. These results uncover two distinct formin polymerase activities. Formins not only promote binding of profilin-actin complexes, but also directly accelerate profilin release from the barbed end via their FH2 domain. These activities are matched to provide a constant rate enhancement over a wide range of profilin-actin concentrations (*Figure 5—figure supplement 1A*). Their combination allows formins to act as pacemakers, which elongate filaments with distinct rates that are buffered against changes in the profilin-actin concentration.

## Formin-mediated actin elongation is resilient to changes in profilin-actin levels

To critically test how our results relate to cellular actin growth, we sought to study actin filament elongation in vivo. Growth of individual actin filaments cannot be visualized in mammalian cells. Formin proteins, however, can be visualized as single molecules in vivo (*Higashida et al., 2004*). We thus established single-molecule TIRFM imaging of constitutively active, mNeonGreen-tagged formins within the cortex of either mammalian mesenchymal (HT1080) or T-lymphocyte (EL4) cells (*Figure 6A–C*). We chose these cell types because of their > 2 fold difference in profilin-actin levels (*Figure 1D*). Because strong overexpression of active formins affects the soluble actin pool (*Dimchev et al., 2017*), we only analyzed cells with extremely low formin levels (Materials and methods). Single formin particles were visible as spots that translocated over μm distances with nearly constant velocity

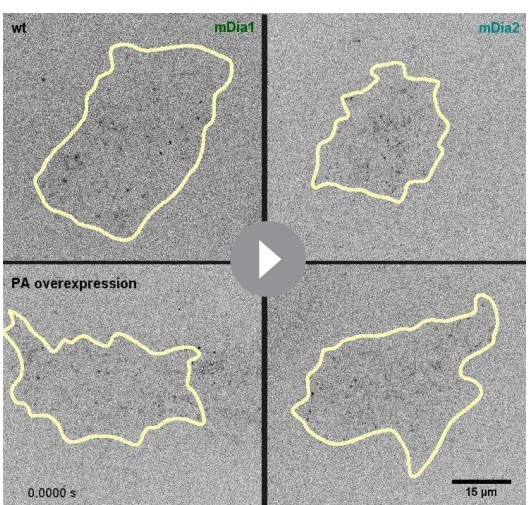

**Video 5.** mDia1 and mDia2 formin single molecule movement in HT1080 cells under conditions with either wt or overexpression of profilin1–actin. mNeonGreen–mDia1/2 FH1-FH2 single molecules were visualized in TIRF-M. To indicate the cell shape, HT1080 cells were masked. Top: mDia1 (left) and mDia2 (right) molecules in wt HT1080 cells. Bottom: mDia1 and mDia2 molecules in HT1080 cells overexpressing profilin and actin.
DOI: https://doi.org/10.7554/eLife.50963.026

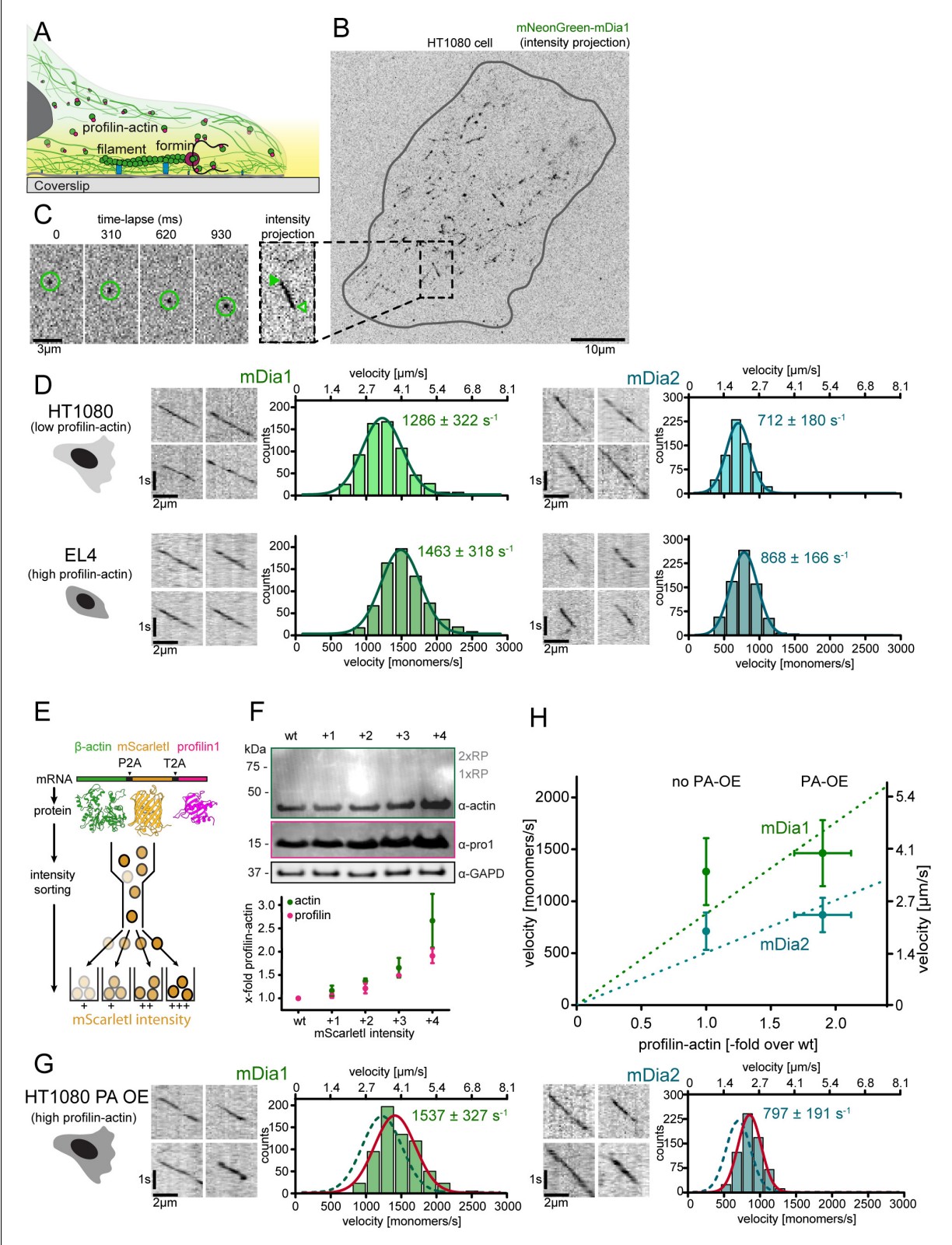

**Figure 6.** Formin single molecule imaging reveals buffered elongation rates in mammalian cells. (**A**) Scheme of TIRFM imaging of single formins in the actin cortex of cells. (**B**) Maximum intensity projection of a TIRFM time-lapse shows growth trajectories of single mNeonGreen-mDia1 molecules in the cortex of a HT1080 cell. Inset: Close-up of a single trajectory as in (**C**). (**C**) TIRFM time-lapse images (left) and intensity projection (right) of an individual mNeonGreen-mDia1 molecule. (**D**) Measurements of mDia1/2 elongation velocities in vivo. Left to right: Scheme of HT1080 (top) and EL4 (lower) cells,

*Figure 6 continued on next page*

*Figure 6 continued*

kymographs of single mNeonGreen- mDia1 (left) or mDia2 (right) molecules followed by velocity distributions. Lines are Gaussian fits. Means and SD are indicated. [$N_{cells} \geq 10$, $n_{molecules/cell} \geq 30$, $n_{total} \geq 650$ per condition]. (E) Workflow to generate profilin1 and β-actin overexpressing HT1080 cells. Polycistronic constructs for β-actin, mScarletI and profilin1 were integrated into the genome. Cells were sorted into four sub-populations dependent on mScarletI fluorescence intensity (*Figure 6—figure supplement 2B*, Materials and methods). (F) Top: Western blot of HT1080 cells (wt or overexpressing sub-populations). No translational read-through is visible (1xRP: actin-mScarletI, 2xRP: actin-mScarletI-profilin1 at expected Mw). Bottom: Relative profilin1 and actin levels (fold over wt) for indicated sub-populations. (G) mDia1/2 velocities in profilin-actin overexpressing HT1080 cells. Left to right: Scheme, kymographs of single mNeonGreen-mDia1 (left) or mDia2 (right) molecules, velocity distributions. Lines are Gaussian fits (Red continuous (PA-OE) and dashed (wt) cells as in (D)). Means and SD are indicated. [$N_{cells} \geq 10$, $n_{molecules/cell} \geq 30$, $n_{total} \geq 650$ per condition]. (H) Mean mDia velocities in HT1080 cells plotted against the relative profilin-actin concentration. Error = SD. Dashed lines are linear fits through the origin.
DOI: https://doi.org/10.7554/eLife.50963.022

The following source data and figure supplements are available for figure 6:

**Source data 1.** Data *Figure 6*.
DOI: https://doi.org/10.7554/eLife.50963.025
**Figure supplement 1.** Control experiments for single formin imaging in vivo.
DOI: https://doi.org/10.7554/eLife.50963.023
**Figure supplement 2.** Profilin1-actin overexpression and in vivo formin speeds at different profilin1-actin levels.
DOI: https://doi.org/10.7554/eLife.50963.024

(*Figure 6B–C*, *Video 5*). Control experiments showed that formin particles corresponded to single molecules (*Figure 6—figure supplement 1*) whose movement was actin polymerization- and not myosin-driven (*Video 6*). Remarkably, we observed that mDia1 and mDia2 moved with distinct speeds that were not only similar between the two cell types (*Figure 6D*), but also strikingly close to their characteristic maximal in vitro velocity (1450 and 820 monomers/s for mDia1 and 2, respectively *Figure 5*).

To test for cell-type-specific regulation as a reason for this invariance, we perturbed profilin-actin levels in a single cell type. Given their low profilin-actin concentration (*Figure 1D*), we overexpressed profilin-actin in HT1080 cells. To prevent side-effects anticipated for the overexpression of profilin alone, we co-overexpressed profilin and actin. To this end, we integrated β-actin with profilin1 and Scarlet-I (as a fluorescent reporter), separated by ribosomal skip sites into a single transgene (*Figure 6E*, Materials and methods). We sorted a heterogeneous pool of stably expressing cells into sub-populations depending on reporter fluorescence (*Figure 6—figure supplement 2B*). Quantification showed that balanced overexpression of profilin and actin levels (2–3-fold) could be achieved in the strongest overexpressing subpopulation (*Figure 6F*, *Figure 6—figure supplement 2A*). Cell fractionation following pharmacological actin arrest (*Peng et al., 2011*) confirmed that profilin remained exclusively in a soluble form, even in this subpopulation (*Figure 6—figure supplement 2C*). Importantly, the soluble actin concentration increased by a very similar amount as the soluble profilin concentration in these overexpressing cells, demonstrating that an approximately 2–3-fold increase in the soluble profilin-actin concentration can be assumed (*Figure 6—figure supplement 2*, see Materials and methods).

We then analyzed the speed of formin-driven actin elongation in these cells. Strikingly, we observed only a marginal increase in mDia1 and mDia2 velocities (by 20% and 12%, respectively) compared to wildtype HT1080 cells (*Figure 6G*). Plotting these velocities against the measured relative profilin-actin levels shows that formin-driven elongation neither strongly nor linearly scaled with profilin-actin concentration (*Figure 6H*). This was also evident when

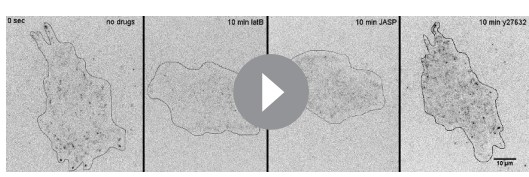

**Video 6.** In vivo mDia2 single molecule movement in absence/presence of latrunculinB, JASP and y27632. To indicate the cell shape, HT1080 cells were masked. mNeonGreen-mDia2 FH1-FH2 single molecules were visualized in TIRF-M. mDia2 molecules were monitored without and after 10 min of drug treatment. The following drugs were applied to the cells, left to right: no drug treatment, 500 nM latrunculinB (latB), 8 μM JASP, 10 μM y27632.
DOI: https://doi.org/10.7554/eLife.50963.027

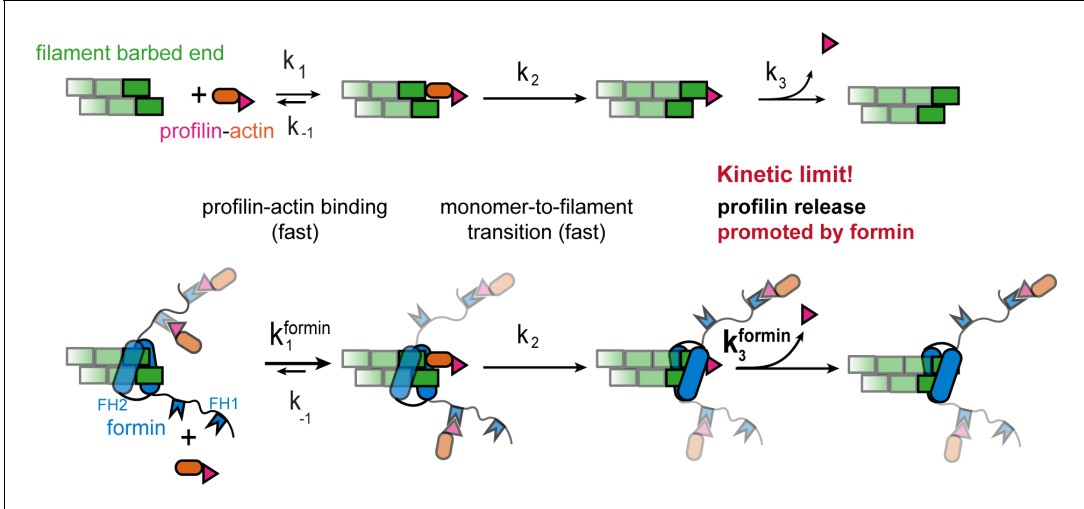

**Figure 7.** Profilin release controls the speed of actin filament growth. Kinetic scheme of the filament elongation cycle from profilin-actin either in the absence (top) or the presence (bottom) of formins. Reaction 1 and 2 are very fast at physiological profilin-actin concentrations, which is why reaction 3 (profilin release from the terminal protomer) kinetically limits the elongation cycle. Formins accelerate both the first and third reaction of the cycle.
DOI: https://doi.org/10.7554/eLife.50963.028

examining formin velocities from both HT1080 and EL4 cells as a function of the various absolute profilin-actin concentration these cells contained. Instead, the data could be well fit with saturation kinetics very similar to those in vitro (*Figure 6—figure supplement 2D*). We conclude that formin-mediated actin elongation in mammalian cells i) is resilient to variations in profilin-actin levels and ii) closely matches the maximal in vitro rates. These findings combined strongly suggest that mammalian cells maintain profilin-actin concentrations near saturation. More importantly, they also indicate that the kinetic limit to actin filament elongation imposed by profilin and formin we discovered in vitro similarly operates in the cytoplasm of living cells.

## Discussion

We have uncovered a biochemical mechanism that kinetically limits actin filament growth at physiological subunit levels (*Figure 7*). Contrary to the textbook model, we demonstrate that actin filament elongation under physiological conditions is not limited by the diffusional encounter between soluble subunits and filament ends. This kinetic mechanism inherently provides robustness to actin dynamics, because it buffers filament growth against changes in the concentration of polymerizable actin in different cellular contexts and across cell types. It is based on two central elements: (i) A bottleneck in filament elongation limiting growth and (ii) maintenance of profilin-actin concentrations near saturation. Both features emerge from the versatile biochemical activities of profilin.

The use of mammalian non-muscle actin in uniquely sensitive assays enabled us to identify the constraints profilin imposes on actin filament growth at physiological conditions. Profilin release is very rapid, but nonetheless kinetically limits filament elongation. The structural monomer-to-filament transition is likely even faster and independent of ATP hydrolysis in actin. This agrees with recent work showing that major structural rearrangements in actin upon polymerization are independent of nucleotide state changes (*Chou and Pollard, 2019*; *Merino et al., 2018*) and with indirect biochemical evidence (*Jégou et al., 2011*). Profilin binds very weakly ($K_D$ >20 μM) to ATP-bound filament barbed ends (*Courtemanche and Pollard, 2013*; *Pernier et al., 2016*) but very tightly to ATP-loaded actin monomers ($K_D$ <0.1 μM). As such, it should out-compete other abundant monomer-binding proteins such as thymosin-$\beta_4$, cofilin, twinfilin and CAP as long as the soluble actin pool is maintained predominantly in a ATP-bound state (*Paavilainen et al., 2004*; *Rosenblatt et al., 1995*). The free profilin concentrations required to simply bind barbed ends from solution are thus unlikely attained in mammalian cells. Our results nonetheless imply that many growing filament ends in cells

are decorated with profilin, however, not as a result of equilibrium binding but through an active, polymerization-coupled mechanism.

Surprisingly, formins stimulate filament growth even at physiological profilin-actin levels. They do so by promoting profilin release through their FH2 domain (*Figure 7*). Profilin and formin appear to mutually destabilize each other at barbed ends, because profilin is known to inhibit formin end-binding (*Pernier et al., 2016*). Structural models suggest that profilin and the formin FH2 domain might directly interfere at barbed ends (*Figure 5—figure supplement 1D*). Alternatively, formins could alter the end structure (*Aydin et al., 2018*) to promote profilin release allosterically. Structures of formin- and profilin-bound barbed ends will be required to resolve this question. Whether and how polymerases unrelated to formins such as Ena/VASP proteins also promote profilin release will be important to study in the future.

Do all actin filaments in mammalian cells grow at their maximal, profilin release-limited speed? The growth speeds we observe are faster than actin network movement in many cellular protrusions (*Renkawitz et al., 2009*), indicating that this is unlikely the case. This mismatch might be explained by filament orientation and, more importantly, compressive forces that slow down filaments pushing against membranes (*Bieling et al., 2016*; *Mueller et al., 2017*). Forces might change the rate-limiting step of filament growth, because they will likely not affect all reactions of elongation equally (*Figure 7*). Brownian ratchet models predict that compression should strongly inhibit the binding of profilin-actin (*Mogilner and Oster, 1996*). The profilin-actin concentration could therefore still affect the growth of filaments experiencing compressive load, such as Arp2/3-generated branched networks that push against cellular membranes and modulate the force these structures generate. This might explain why rapidly moving cells contain higher profilin-actin levels.

Competition for soluble actin has been proposed (*Suarez and Kovar, 2016*) to explain mutual inhibition between distinct actin structures in cells (*Rotty et al., 2015*; *Suarez et al., 2015*). How can this be reconciled with robust actin growth from a large profilin-actin pool we observe? Mutual inhibition might not necessarily originate from direct competition for a limiting monomer resource, but indirect effects that can be understood when considering the partitioning of actin monomers into distinct soluble and filamentous forms: The disassembly of entire classes of actin networks by either genetic or pharmacological inhibition of actin nucleators such as the Arp2/3 complex or formins will transiently mobilize and liberate actin monomers usually contained in filaments. Because the cellular profilin pool is finite and lower than the total actin concentration, such transient disassembly might exceed profilin's capacity to bind and soak up actin monomers. Because actin nucleation -both catalyzed and spontaneous- is strongly promoted by free actin monomers, this might in turn trigger nucleation through the remaining, unperturbed nucleation pathways resulting in homeostatic filament amounts. Such a monomer-triggered mechanism has been proposed for formin-mediated nucleation after cell deformation (*Higashida et al., 2013*). Ways to detect distinct soluble actin states in vivo are needed to understand their effect on local actin network dynamics (*Skruber et al., 2018*).

# Materials and methods

**Key resources table**

| Reagent type (species) or resource | Designation | Source or reference | Identifiers | Additional information |
|---|---|---|---|---|
| Strain, strain background (*Escherichia coli*) | BL21 Star pRARE | EMBL Protein Expression Facility | | Chemically competent cells |
| Strain, strain background (*Escherichia coli*) | BL21 Rosetta | Novagen | Cat# 70954 | Chemically competent cells |
| Cell line (*S. frugiperda*) | SF9 | A. Musacchio, MPI Dortmund | RRID:CVCL_0549 | Cell line for virus generation |

*Continued on next page*

*Continued*

| Reagent type (species) or resource | Designation | Source or reference | Identifiers | Additional information |
|---|---|---|---|---|
| Cell line (*T. ni*) | TnaO38 | A. Musacchio, MPI Dortmund | RRID:CVCL_Z252 | Cell line for protein expression from baculovirus system |
| Cell line (*Homo-sapiens*) | HT1080 | ATCC | Cat# CCL-121, RRID:CVCL_0317 | Profilin and actin quantifications by WB, formin single molecule transfection |
| Cell line (*Homo-sapiens*) | B16F10 | ATCC | Cat# CRL-6475, RRID:CVCL_0159 | Profilin and actin quantifications by WB, formin single molecule transfection |
| Cell line (*Homo-sapiens*) | BMDC | Lab of M. Piel, Institut Curie, Paris | | Profilin and actin quantifications by WB |
| Cell line (*M. musculus*) | neutrophils | Lab of M. Piel, Institut Curie, Paris | | Profilin and actin quantifications by WB |
| Cell line (*M. musculus*) | EL4, T-lymphocytes | Lab of M. Taylor, MPI Berlin | | Profilin and actin quantifications by WB |
| Transfected construct | pΔCMV-mNeongreen-mDia1FH1-2 | This paper | Uniprot: O08808 | transfected construct, can be obtained in the lab of P. Bieling, MPI Dortmund |
| Transfected construct | pΔCMV-mNeongreen-mDia2FH1-2 | This paper | Uniprot: Q9Z207 | transfected construct, can be obtained in the lab of P. Bieling, MPI Dortmund |
| Transfected construct | pPBCAG-β-actin-P2A-mScarletI-T2A-profilin1 | This paper | | transfected construct, can be obtained in the lab of P. Bieling, MPI Dortmund |
| Antibody | anti-actin (mouse monoclonal) | ThermoFisher | Cat# MA5-11869, RRID:AB_11004139 | WB (1:1000) |
| Antibody | anti-profilin1 (mouse monoclonal) | Sigma Aldrich | Cat# 061M4892 | WB (1:20000) |
| Antibody | anti-profilin2 (mouse monoclonal) | Santa Cruz | Cat# sc-100955, RRID:AB_2163221 | WB (1:20000) |
| Antibody | anti-GAPDH (14C10) (rabbit monoclonal) | Cell Signaling | Cat# 2118, RRID:AB_561053 | WB (1:5000) |
| Antibody | anti-mouse (donkey polyclonal) | Licor | Cat# 925–32212, RRID:AB_2716622 | WB (1:10000) |
| Antibody | anti-rabbit (donkey polyclonal) | Licor | Cat# 926–68073, RRID:AB_10954442 | WB (1:10000) |
| Recombinant DNA reagent | pFL-h.s. β-actin_wt-linker-T4b (plasmid) | This paper | Uniprot: P60709 | β-actin insect cell expression, can be obtained in the lab of P. Bieling, MPI Dortmund |
| Recombinant DNA reagent | pFL-h.s. β-actin_Q137A_D154A_H161A-linker-T4b (plasmid) | This paper | | β-actin insect cell expression, can be obtained in the lab of P. Bieling, MPI Dortmund |
| Peptide, recombinant protein | Streptavidin | Sigma Aldrich | Cat. #: 189730 | For filament attachment |
| Chemical compound, drug | Latrunculin B | Sigma Aldrich | Cat. #: L5288 | For actin arrest |

*Continued on next page*

*Continued*

| Reagent type (species) or resource | Designation | Source or reference | Identifiers | Additional information |
| --- | --- | --- | --- | --- |
| Chemical compound, drug | Y-27632 dihydrochloride | Sigma Aldrich | Cat. #: Y0503 | For actin arrest |
| Chemical compound, drug | jasplakinolide | Sigma Aldrich | Cat. #: J4580 | For actin arrest |
| Chemical compound, drug | phalloidin | Sigma Aldrich | Cat. #: P2141 | For actin arrest |
| Chemical compound, drug | 1,5-IAEDANS | Thermo Fisher | Cat. #: I14 | For actin labeling |
| Chemical compound, drug | EZ-Link Maleimide-PEG2-Biotin | Thermo Fisher | Cat. #: A39261 | For actin labeling |
| Chemical compound, drug | $\gamma-^{32}P-ATP$ (3000 Ci/mmol) | PerkinElmer | Cat. #: NEG002A | For ATPase assay |
| Chemical compound, drug | HO-PEG-NH$_2$ and Biotin-CONH-PEG-O-C$_3$-H$_6$-CONHS | Rapp Polymere | # 103000–20 and # 133000-25-35 | For glass surface functionalization |

## Structural models of barbed end complexes

Using MODELLER (*Webb and Sali, 2016*) we built models of the binding of profilin, formin or profilin-formin to the barbed end of the actin filament. For the profilin models we superimposed the actin monomer in the profilin/β-actin crystal structure (PDBID 2BTF) (*Schutt et al., 1993*) to either the ultimate or penultimate subunit of a filament barbed end. As a filament template we used our recent structure of α-actin in complex with beryllium fluoride (PDBID 5OOF) (*Merino et al., 2018*).

To model the FH2 domain of mDia1 (aa 750–1163) bound to a profilin-occupied barbed end we superimposed subdomains 1 (aa 1–33; 70–137; 348–375) and 3 (aa 138–180; 274–347) of an actin subunit from the Bni1p-actin crystal structure (PDBID 1Y64) (*Otomo et al., 2005*) with the terminal monomers in F-actin-BeFx. This brings the FH2 domain of the formin to the right position in the actin filament. Given that the Bni1p structure has a non-physiological helical arrangement of the formin, we erased the loop between its Knob and Lasso regions (aa 804–831 in mDia1) and built it de novo to recover the known dimeric arrangement of the FH2 domains. To further improve the quality of the models we also included two crystal structures of the FH2 domains of mDia1 (PDBID 1V9D and 3O4X) (*Nezami et al., 2010*; *Shimada et al., 2004*).

## Protein design

We used the RossetaScripts framework (*Fleishman et al., 2011*) within Rosetta (*Leaver-Fay et al., 2011*) to find possible mutations to increase the affinity of profilin for actin. For the design we tested a model built on the crystal structure of profilin-β-actin (*Schutt et al., 1993*) as well as our F-actin-profilin models (see previous section). The design strategy was modified from the protocol provided by the Baker lab in *Berger et al. (2016)* (see Computational methods: design with ROSETTA in their manuscript). We tried mutating all profilin residues at the interface with actin, but did not allow mutations into Cys, Pro, Trp, or Gly. We generated a total of 1920 possible profilin sequences for each actin conformation, and kept the top 50 (lowest energies) for further analysis. From there, we selected single mutations likely to increase the affinity of profilin for actin and tested them experimentally.

## Protein purification and labeling

### 10xhis-Gelsolin G4-6

Mouse Gelsolin G4-6 was cloned with an N-terminal 10xhis tag into a pCOLD vector. Protein was expressed in *E. coli* BL21 Rosetta cells for 16 hr at 16°C. After cell lysis (20 mM Tris-Cl pH 8.0, 300 mM KCl, 5 mM CaCl$_2$, 0.2 mM ATP, 0.5 mM β-mercaptoethanol, 1 mM PMSF, DNAseI) the lysate was hard spun and purified by IMAC over a 40 ml Ni$^{2+}$ superflow column. Protein was gradient eluted (20 mM Tris-Cl pH 8.0, 300 mM KCl, 5 mM CaCl$_2$, 0.2 mM ATP, 500 mM Imidazole) over 10

column volumes followed by gelfiltration over Superdex 200 26/600 into storage buffer (5 mM Tris-Cl pH 8.0, 50 mM KCl, 5 mM $CaCl_2$, 0.1 mM ATP, 0.5 mM TCEP, 20% glycerol). The protein was snap frozen in liquid nitrogen and placed in −80°C for long-term storage.

## Native bovine (β, γ)-actin

Bovine thymus was manually severed into small fragments and mixed in a precooled blender together with ice cold Holo-Extraction buffer (10 mM Tris-Cl pH 8.0, 7.5 mM CaCl2, 1 mM ATP, 5 mM β-mercaptoethanol, 0.03 mg/ml benzamidine, 1 mM PMSF, 0.04 mg/ml trypsin inhibitor, 0.02 mg/ml leupeptin, 0.01 mg/ml pepstatin, 0.01 mg/ml apoprotein). After homogenizing, additional 2.5 mM β-mercaptoethanol was added to the lysate and the pH was checked and readjusted to pH 8.0 if necessary. After initial centrifugation the lysate was filtered through a nylon membrane [100 μm] and hard spun in an ultracentrifuge. The volume of the cleared supernatant was measured out and the salt and the imidazole concentrations were adjusted (KCl to 50 mM, imidazole to 20 mM). The supernatant was incubated with the gelsolin G4-6 fragment to promote the formation of actin:gelsolin G4-6 complexes. To this end, 4 mg of 10xhis-gelsolinG4-6 were added for each g of thymus to the lysate and dialyzed into IMAC wash buffer overnight (10 mM Tris-Cl pH 8.0, 50 mM KCl, 20 mM imidazole, 5 mM $CaCl_2$, 0.15 mM ATP, 5 mM β-mercaptoethanol). The lysate containing the actin:gelsolin G4-6 complex was then circulated over a $Ni^{2+}$ superflow column. Actin monomers were eluted with Elution Buffer (10 mM Tris-Cl pH 8.0, 50 mM KCl, 20 mM imidazole, 5 mM EGTA, 0.15 mM ATP, 5 mM β-mercaptoethanol) into a collection tray containing $MgCl_2$ (2 mM final concentration). Actin containing fractions were identified by gelation, pooled and further polymerized for 4 hr at RT after adjusting to 1xKMEI and 0.5 mM ATP. After ultracentrifugation, the actin filament pellet was resuspended in F buffer (1xKMEI, 1xBufferA) and stored in continuous dialysis at 4°C. F buffer containing fresh ATP and TCEP was continuously exchanged every 4 weeks.

For fluorescence measurements actin monomers were labeled with 1.5-IAEDANS at Cys374 as outlined in *Hudson and Weber (1973)*; *Miki et al. (1987)* using a modified protocol. Briefly, the actin filament solution was transferred to RT, mixed with 10x molar excess of 1.5-IAEDANS and incubated for 1 hr at RT. The reaction was quenched by the addition of 1 mM DTT for 10 min. After ultracentrifugation at 500.000xg for 30 min, the actin pellet was resuspended in an appropriate amount of BufferA and dialyzed in the same buffer at 4°C for 2 days. Actin monomers were separated from residual filaments by centrifugation at 300.000xg followed by determination of monomer concentration and degree of labeling at 280 nm/336 nm.

## Recombinant human β-actin

Human β-actin was cloned with a C-terminal linker sequence (ASRGGSGGSSGGSA) followed by the human β-thymosin sequence followed by a 10xhis tag (*Noguchi et al., 2007*) in a pFL vector. PCR-based site directed mutagenesis was performed to generate human, ATPase deficient β-actin (Q137A+D154A+H161A). Proteins were expressed in insect TnaO38 cells for 3 days at 27°C. The cells were resuspended with a 5x pellet volume of lysis buffer (10 mM Tris-Cl pH 8.0, 50 mM KCl, 7.5 mM $CaCl_2$, 1 mM ATP, 5 mM imidazole, 5 mM β-mercaptoethanol, 0.03 mg/ml benzamidine, 1 mM PMSF, 1x complete protease inhibitor cocktail). After cell lysis by a microfluidizer the lysate was hard spun, filtered through a 0.45 μm syringe filter and passed through a $Ni^{2+}$-sepharose excel column. After washing the column with 10 mM Tris-Cl pH 8.0, 50 mM KCl, 5 mM $CaCl_2$, 0.15 mM ATP, 5 mM imidazole, 5 mM β-mercaptoethanol, the protein was eluted over a 6 CV linear gradient to Elution Buffer (10 mM Tris-Cl pH 8.0, 50 mM KCl, 0.15 mM ATP, 300 mM imidazole, 5 mM β-mercaptoethanol) followed by dialysis into BufferA overnight. Next, the protein was cleaved with TLCK-treated chymotrypsin in a molar ratio of 250:1 (actin over chymotrypsin) at 25°C. After 10 min the reaction was quenched with 0.2 mM PMSF at 4°C. The protein was again passed over the $Ni^{2+}$-sepharose excel column and the flow through was polymerized for 3 hr at 25°C by the addition of 1xKMEI, 2 mM $MgCl_2$ and 0.5 mM ATP. After hard spin, the actin filament pellet was resuspended into F buffer (1xKMEI, 1xBufferA) and stored in dialysis at 4°C.

## Profilin 1 and 2

Human profilin isoforms 1 and 2 were expressed either as untagged proteins or with an N-terminal SUMO3-10xhis tag in *E. coli* BL21 Rosetta cells at 30°C for 4.5 hr. Profilin1 mutants that were

generated via site directed mutagenesis (E82A, R88K, K125E E129K and S71M) were expressed with an N-terminal SUMO3-10xhis in *E. coli* BL21 Rosetta cells at 30°C for 4.5 hr. For the N-terminal SUMO3-10xhis tagged version, the cells were lysed (20 mM Tris-Cl pH 8.0, 300 mM NaCl, 10 mM imidazole, 0.5 mM β-mercaptoethanol, 15 μg/ml benzamidine, 1 mM PMSF and DNaseI) and the hard spun lysate was circulated over a 5 ml HiTrap Chelating column followed by overnight SenP2 cleavage of the N-terminal SUMO-his tag on the column, generating the natural profilin N-terminus. After cleavage the flow through was gelfiltered over a Superdex 200 16/600 column into storage buffer (20 mM Tris-Cl pH 7.5, 50 mM NaCl, 0.5 mM TCEP). The non-tagged profilin isoforms were purified as described in *Bieling et al. (2018)* by ammonium sulfate precipitation, followed by ion-exchange (DEAE) and hydroxylapatite (HA) chromatography steps, followed by size exclusion chromatography (Superdex 200 16/600) into storage buffer (20 mM Tris-Cl pH 7.5, 50 mM NaCl, 0.5 mM TCEP). Proteins were snap frozen in liquid nitrogen with the addition of 20% glycerol in the storage buffer and were stored at −80°C.

## Profilin - Actin complex
Filamentous mammalian actin was depolymerized through dialysis into BufferA (2 mM Tris, 0.2 mM ATP, 0.1 mM $CaCl_2$, 0.1 μg/ml $NaN_3$, 0.5 mM TCEP) and gelfiltered over a Superdex 200 16/600. After gelfiltration a 1.5x molar excess of profilin was added to the actin monomers and incubated at 4°C overnight to form profilin-actin complexes. Profilin-actin was then separated from excess free profilin by gelfiltration over a Superdex 200 10/300 GL into BufferA. The complex was concentrated to working concentrations between 200 and 400 μM and stored at 4°C up to two weeks without inducing nucleation.

## Formins
*M. musculus* mDia1 FH1-2 (aa 548–1154), mDia2 FH1-2 (aa 515–1013), $FH1_{mdia1}FH2_{mdia2}$ (aa 548-751/453-1013), $FH1_{mdia2}FH2_{mdia1}$ (aa 515-612/645-1154), *H. sapiens* DAAM1 FH1-2 (aa 490–1029), were expressed with an N-terminal 10xhis-SNAP-tag. All constructs were expressed in *E. coli* BL21 Star pRARE cells for 16 hr at 18°C. The cells were lysed in Lysis Buffer (50 mM $NaPO_4$ pH 8.0 (pH 7.5 for mDia chimera constructs), 400 mM NaCl, 0.75 mM β-mercaptoethanol, 15 μg/ml benzamidine, 1xcomplete protease inhibitors, 1 mM PMSF, DNaseI,) and the protein was purified by IMAC using a 5 ml HiTrap column. The protein was eluted using Elution Buffer (50 mM $NaPO_4$ pH 7.5, 400 mM NaCl, 400 mM imidazole, 0.5 mM β-mercaptoethanol) in a gradient and the 10xhis-tag was directly cleaved using TEV protease overnight. After cleavage proteins were desalted into low salt Mono S buffer (10 mM Hepes pH 7.0 (pH 7.5 for mDia chimera constructs), 90 mM NaCl, 0.5 mM TCEP) over a HiPrep 26/10 desalting column followed loading onto a MonoS column. Protein was eluted by a linear 25 column volume gradient to high salt MonoS buffer (10 mM Hepes pH 7.5, 1 M NaCl, 0.5 mM TCEP) followed by gelfiltration over a Superdex 200 16/600 into storage buffer (20 mM Hepes pH 7.5, 200 mM NaCl, 0.5 mM TCEP, 20% glycerol).

Following the purification the proteins were either snap frozen and stored in –80°C or directly used for SNAP-labeling. A 3x molar excess of SNAP Cell TMR-star was mixed with the protein and incubated for 6 hr at 16°C followed by an overnight incubation on ice. Post labeling the protein was gelfiltered over a Superose 6 10/300 GL column into storage buffer. The degree of labeling (50–70%) was determined by absorbance at 280 nm and 554 nm.

## Myosin and biotinylated heavy - mero – myosin (HMM)
Skeletal muscle myosin was prepared from chicken according to *Pollard (1982)*. Briefly, 300 g muscle tissue were mixed with 4x volumes of extraction buffer (0.15 mM $KH_2PO_4$ pH 6.5, 0.3 M KCl, 5 mM $MgCl_2$, 0.1 mM ATP, 20 mM EDTA) while blending. The pH was adjusted to 6.6 afterwards. After centrifugation, the supernatant was diluted with 10x volumes of cold water and the precipitate was separated from solution by centrifugation at 9.000xg for 30 min. The pellet was resuspended in buffer 8 (3 ml buffer per g of pellet, 60 mM $KH_2PO_4$ pH 6.5, 1 M KCl, 25 mM EDTA) and dialyzed against buffer 9 (25 mM $KH_2PO_4$ pH 6.5, 0.6 M KCl, 10 mM EDTA, 1 mM DTT) over night. Following dialysis, an equal volume of cold water was added to the myosin solution and stirred for 30 min. After centrifugation for 30 min at 15.000xg, the supernatant was diluted with 7 volumes of cold water and again spun for 30 min at 9.000xg. The pellet fraction was then resuspended into buffer 10

(20 mM Tris-Cl pH 7.0, 0.6 M KCl, 10 mM DTT) and treated with α-chymotrypsin (25 µg/ml final) at 25°C for 15 min. The reaction was quenched by the addition of 0.3 mM PMSF. After protease treatment, the myosin was dialyzed into buffer 11 (10 mM NaPi pH 7.2, 35 mM NaCl, 10 mM DTT). On the next day, the HMM was separated by ultra-centrifugation for 1 hr at 300.000xg. The supernatant was desalted into buffer 11 without DTT and incubated with 15x molar excess of EZ-Link maleimide-PEG11-biotin for 2 hr on ice. The reaction was stopped by the addition of 1 mM DTT. The protein was desalted into buffer 11 containing 20% glycerol, SNAP-frozen and stored at –80°C.

## Biochemical assays

### Buffers

All biochemical experiments were carried out in a common final assay buffer of the following composition if not stated otherwise: 20 mM Hepes pH 7.0, 100 mM KCl, 1.5 mM MgCl$_2$, 1 mM EGTA, 20 mM β-mercaptoethanol, 0.1 mg/ml β-casein, 1 mM ATP. This buffer has a molar ionic strength of 0.133 M, which is close to the physiological ionic strength found in literature (between 0.1 and 0.2 M).

## Endpoint hydrolysis measurements via HPLC

All HPLC measurements were initiated by loading actin monomers and profilin-actin with Mg-ATP. After a 1 hr incubation of monomers and profilin-actin (40 µM) with 1 mM MgCl$_2$ and 1 mM ATP, proteins were desalted into 2 mM Tris-Cl pH 8.0 using a Zeba Spin Desalting column. Actin seeds were then polymerized from the desalted actin monomers by adjusting to 1xKMI (50 mM KCl pH 7.0, 1.5 mM MgCl$_2$, 10 mM imidazole) for 1 hr at 23°C. To start the reaction, profilin–actin (40 µM) was mixed with seeds (5 µM) in presence of 1xKMI. After 1.5 hr incubation at 23°C, the samples were boiled for 5 min followed by a hard spin. The supernatant was carefully aspired and analyzed by HPLC. As a negative control, profilin-actin were stabilized with 5 mM latrunculin B and the seeds were incubated with 5 mM phalloidin before mixing, otherwise the samples were treated as mentioned above.

All nucleotide retention times were measured using an UltiMate 3000 HPLC Dionex – System. The samples were injected onto a C18-column equilibrated with 16% acetonitrile, 50 mM KP$_i$ pH 6.6, 10 mM TBABr. The nucleotide signal intensity was recorded at 254 nm.

## Radioactive ATPase assays

100 µM Mg-ATP-actin was dialyzed into BufferA for 7 days. After gelfiltration over a Superdex200 16/60 the actin monomer fraction was split into two fractions. With the addition of 1.5x-molar excess profilin1 to one of the monomer fractions, profilin-actin complexes were formed and isolated over a Superdex200 10/300 GL. Both actin monomer and profilin-actin fractions were desalted into ATP free BufferA (2 mM Tris-Cl pH 8.0) over a Zeba Desalting column. 1 ml of 10 µM actin monomers was incubated with 2xKMEI to polymerize actin for 1 hr at RT. In the meantime, 1 ml of 10 µM profilin-actin was incubated with 0.2 mM EGTA, 0.132 mM MgCl$_2$ and 0.06 mM γ–$^{32}$P–ATP (3000 Ci/mmol, PerkinElmer #NEG002A) for 30 min on ice. After incubation, γ– $^{32}$P–ATP labeled profilin-actin complexes were desalted over a Zeba Desalting column into 2 mM Tris–Cl pH 8.0, 0.2 mM EGTA, 0.132 mM MgCl$_2$. Immediately before introducing the pre-polymerized actin seeds into the experiments, seeds were sheared through a 27 G needle. The ATPase assay reaction was started by rapidly mixing 6 µM of actin seeds with 6 µM of γ–$^{32}$P–ATP labeled profilin-actin. 100 µl samples were taken at different time points over a time course of 48 min and immediately quenched with an equal volume of silicotungstic–sulfuric acid (4.3% aqueous silicotungstic acid in 2.8 N sulfuric acid). Samples were recovered in 1 ml of a 1:1 isobutanol/xylene solution and immediately rigorously mixed with additional 100 µl of 10% ammonium molybdate for 20 s. After 4 min centrifugation at 200xg the upper phase containing the phosphate molybdate complex was extracted. The complex was diluted in LSC cocktail (Hidex) and the number of counts was detected using a liquid scintillation counter (Triathler multilabel tester, Hidex).

## Fluorescence anisotropy experiments

The measurements were performed in 96 well CORNING plates with a TECAN SPARK plate reader. A constant concentration of 150 nM (for wt and weakly binding profilin) or 40 nM (for tightly binding profilin) actin monomers were stabilized with 25 µM latrunculin B and mixed with 4 nM Atto488-

WAVE1(WCA) (*Bieling et al., 2018*). Profilin was titrated to the Atto488-WAVE1(WCA):actin complex to final concentrations of 0–20 µM and equilibrated for 5 min at RT before the measurement. The assay was performed in 1xTIRF buffer (20 mM Hepes pH 7.0, 100 mM KCl, 1.5 mM MgCl$_2$, 1 mM EGTA, 20 mM β-mercaptoethanol, 0.1 mg/ml β-casein, 1 mM ATP). For the determination of anisotropy values, Atto488-WAVE1(WCA) was excited at 485/20 nm and the emission was detected at 535/25 nm.

## IAEDANS fluorescence quenching measurements

Fluorescence measurements were performed in 96 well CORNING plates with a TECAN SPARK plate reader. A constant concentration of 150 nM 1.5-IAEDANS labeled actin monomers were premixed with 25 µM latrunculin B in 1xTIRF assay buffer and thymosin-β$_4$ was titrated over a range of 0–200 µM. The 1.5-IAEDANS actin was excited at 336 nm and the emission and thus the fluorescence change of the 1.5-IAEDANS actin bound to thymosin-β$_4$ was detected at 490 nm.

## Tryptophan fluorescence quenching by stopped flow

To determine the association rate constant for profilin binding to actin monomers, increasing profilin concentrations were mixed in a 1:1 vol with a fixed concentration of 0.5 µM actin monomers at 25°C. The assay was performed in 20 mM Hepes pH 7.0, 100 mM KCl, 1.5 mM MgCl$_2$, 1 mM EGTA, 20 mM β-mercaptoethanol, 1 mM ATP, 1.5 µM latrunculin B. Tryptophan fluorescence intensity was recorded by a SX20 double mixing stopped flow device (Photophysics) using excitation and emission wavelengths of 280 and 320 nm, respectively. The time courses of tryptophan fluorescence was recorded and fitted with a single exponential function to yield the observed pseudo-first order reaction rate ($k_{obs}$) as a function of profilin concentration.

## Single filament experiments on functionalized glass coverslips using TIRF-Microscopy

Flow chambers were prepared from microscopy counter slides passivated with PLL-PEG and coverslips (22 × 22 mm, 1.5 hr, Marienfeld-Superior) that were functionalized according to *Bieling et al. (2016)*. Briefly, coverslips were cleaned with 3 M NaOH and Piranha solution followed by silanization and PEG-biotin/hydroxy functionalization. For the single filament assays the flow cell surfaces were blocked for 5 min with a Pluronic block solution (0.1 mg/ml κ-Casein, 1% Pluronic F-127, 1 mM TCEP, 1xKMEI), followed by 2 washes with 40 µl of wash buffer (0.5 mM ATP, 1 mM TCEP, 1xKMEI, 0.1 mg/ml β-Casein). The channel was incubated with 75 nM streptavidin for 3 min, followed by washing and incubation of 90 nM biotin-phalloidin for 3 min. Pre-polymerized actin seeds were immobilized in the channel for another 2 min for cases when spontaneous nucleation was not rapid enough (e.g. low profilin-actin concentrations, absence of formins).

Visualization by TIRF-M was performed following a modified protocol as outlined in *Hansen and Mullins (2010)* and *Kuhn and Pollard (2005)*. Briefly, 9 µl of a 4.44x µM profilin-actin solution was mixed with 1 µl of 10x ME (0.5 mM MgCl$_2$, 2 mM EGTA) and 4 µl oxygen scavenging system (1.25 mg/ml glucose-oxidase, 0.2 mg/ml catalase, 400 mM glucose) (*Aitken et al., 2008*; *Bieling et al., 2010*; *Rasnik et al., 2006*). The Mg-ATP–profilin-actin was then combined with 26 µl reaction buffer mix containing additives including 10 nM Cy5-UTRN$_{261}$, (plus additives as described in the specific results section and in the corresponding figure legends) and TIRF buffer with the final composition of: 20 mM Hepes pH 7.0, 100 mM KCl, 1.5 mM MgCl$_2$, 1 mM EGTA, 20 mM β-mercaptoethanol, 0.1 mg/ml β-casein, 0.2% methylcellulose (cP400, M0262, Sigma-Aldrich), 1 mM ATP and 2 mM Trolox.

Filaments that appeared to either stop growing due to surface defects or that showed very large movements out of the TIRF field were not analyzed. All single filament polymerization experiments were performed using profilin-actin as a substrate unless otherwise indicated in the figure legends.

## Microfluidic single filament experiments by TIRF microscopy

Experiments were essentially conducted as described in the previous section with the following modifications: Microfludic PDMS chambers were mounted on PEG – biotinylated glass cover slips via plasma treatment as described in *Duellberg et al. (2016)*. The chambers were designed with 2 or 3 inlets and one observation channel. After pluronic block (0.1 mg/ml κ-Casein, 1% Pluronic F-127, 1 mM TCEP, 1xKMEI) for 5 min, biotinylated Alexa647-phalloidin stabilized actin seeds were bound to

the surface via streptavidin. To start actin filament polymerization, profilin-actin was diluted in TIRF buffer and directly transferred from a syringe pump into the reaction chamber to visualize filament elongation immediately under the TIRF-microscope. The flow speed was set to 14–16 µl/min.

## TIRF-Microscopy data acquisition

All in vitro experiments were performed at RT using a custom built TIRF microscope (OLYMPUS IX81). Image acquisition was done by a EM CCD Andor iXon 888 camera controlled by Micromanager 1.4 software (*Edelstein et al., 2014*). Fiji ImageJ was used for image and data analysis. Dual color imaging was performed through a 60x OLYMPUS APO N TIRF objective using TOPTICA IBeam smart 640 s and 488 s/or OBIS 561 nm LS lasers and a Quad-Notch filter (400-410/488/561/631-640). Shutters, optical filters, dichroic mirrors and the Andor camera were controlled by Micromanager 1.4 software (*Edelstein et al., 2014*). Images were acquired between intervals of 0.14–10 s using exposure times of 30–200 ms to avoid bleaching.

All in vivo single molecule experiments were performed at 23°C unless otherwise specified using a customized Nikon TIRF Ti2 microscope and Nikon perfect focus system. Image acquisition was achieved by dual camera EM CCD Andor iXon system (Cairn) controlled by NIS – Elements software. Dual color imaging was performed through an Apo TIRF 60x oil DIC N2 objective using a custom multilaser launch system (AcalBFi LC) at 488 nm and 560 nm. Images were acquired at intervals of 0.075–0.15 s.

## Cell culture

HT1080 cells were cultured in DMEM and supplemented with 2 mM glutamine, 1% NEAA and 10% FBS. B16F10 cells were cultured in DMEM and supplemented with 4 mM glutamine, 1% NEAA and 10% FBS. Mouse EL4 cells were cultured in RPMI-1640 with 10% FBS. The cells were cultivated at 37°C with 5% $CO_2$ in a humidified incubator. BMDCs were cultured according to *Vargas et al. (2016)*. Mouse neutrophil cells were extracted from mouse blood. The identity has been authenticated by STR profiling. All cell lines were tested negative for mycoplasma contamination.

## Quantitative western blot analysis

Quantitative western blots were performed using 12% SDS gels. To determine actin and profilin amounts per cell, purified actin and profilin references of known concentration were titrated into 1xPBS on the same gel as the cell lysate samples. The number of cells was counted by a Vi-CELL Viability Analyzer from Beckmann Coulter. Cells were lysed in 5 mM Tris–Cl pH 7.5, 150 mM NaCl, 1 mM EDTA, 1% Triton X-100 and 10 min of sonication. All protein samples were prepared in 1x Laemmli sample loading buffer (Cold Spring Harbor Protocols, 2007). Precision Plus Protein Standard All Blue (Biorad) was used as a molecular weight marker. SDS Gel electrophoresis was performed in Tris-Glycine buffer and proteins were transferred onto a PVDF membrane (Merck Chemicals). After protein transfer membranes were blocked with Odyssey TBS blocking solution (LI-COR Biosciences) for 1 hr at RT and probed with one of the following antibodies: monoclonal mouse anti – actin (1:1000, #MA5-11869 ThermoFisher)/profilin1 (1:20000, #061M4892 **Sigma**)/profilin2 (1:20000, #sc-100955 Santa Cruz) and monoclonal rabbit anti - GAPDH(14C10) (1:5000, #2118) as primary antibodies. As secondary antibody infrared labeled - donkey anti-mouse and donkey anti-rabbit were used (1:10000, #925–32212, #926–68073 LI-iCOR). All antibodies were incubated for 1 hr at RT and the membrane was washed with TBS-T (TBS + 0.05% Tween20) in between. The antibody signal was visualized by fluorescence detection on a LI-COR Odyssey CLx imaging system.

## Cell volume measurements fluorescence eXclusion

Cell volumes were determined for different cell lines and primary cells as outlined in the text. Measurements were performed as described in *Cadart et al. (2017)* for all cell types in suspension or attached to a glass surface using fibronectin.

## Single molecule visualization of formins in cells

Constitutively active fragments of mDia1 FH1-2 (aa 548–1154) and mDia2 FH1-2 (aa 515–1013) were cloned with an N-terminal mNeonGreen sequence in a pΔCMV vector.

20.000 cells of HT1080 were seeded into a well of an eight well Lab-Tek 1.5H that was coated with fibronectin (40 µg/ml). On the next day, 1.5 µl FuGENE (Promega) were incubated in 150 µl OptiMEM (Gibco) for 5 min at 23℃ followed by a 15 min incubation with 0.5 µg DNA. The entire transfection mix was directly transferred to the cells.

$2 \times 10^6$ EL4 cells were resuspended in 100 µl Nucleofector solution and 2 µg DNA and electroporated by Lonza Amaxa NUcleofector II with the appropriate program. After electroporation, the cells were transferred into 1.5 ml medium. To minimize the transfer of cell debris, cells were once passaged on the following day. Finally, the cells were seeded onto a mouse ICAM-1 coated Lab-Tek 1.5H.

For either cell type after 18 hr after transfection (HT1080) or initial passage (EL4), the cell culture medium was replaced by HBSS (PAN Biotech #P04-32505). To obtain a more direct comparison with our in vitro measurements, which were carried out at room temperature, we imaged cells at room temperature quickly after transferring them to the microscope. Only cells with very low formin expression (<35 molecules per cell per image) were chosen for image acquisition. To prevent an influence of mechanical resistance on formin movement, we only analyzed molecules that translocated freely in the interior of the cell and did not get close the cell periphery, where their movement might be obstructed by the plasma membrane.

Control experiments were performed incubating the cells with either 500 nM latrunculin B, 10 µM Y-27632 or 8 µM JASP (*Peng et al., 2011*). Imaging was performed either immediately before or 10 min after drug treatment.

## Overexpression of profilin1 and β - actin in HT1080 cells

Polyclonal HT1080 cell lines were generated using the PiggyBac system according to System Bioscience protocols. For profilin-actin overexpression, the following sequences were cloned in a pBP-CAG vector: human β-actin–P2A–mScaletl–T2A–human profilin1 via Gibson assembly.

After transfection of a construct containing the sequence: actin-P2A-mScarletl-T2A-profilin1, transgenetic cells were selected using puromycin (1 µg/ml) followed by cell sorting through a flow cytometer (BD FACSAria). The distinct sub-populations of the cells were sorted according to their fluorescence intensity and then grown separately. Quantitative western blot analysis was performed to measure the profilin1 and β-actin amounts in these distinct cell populations. We did not detect any actin-containing proteins of larger molecular weight that could potentially result from ribosomal read-through (*Figure 6F*), presumably because of actin's stringent folding requirements.

### Quantification and statistical data analysis

All analyzed data were plotted and fitted in Origin9.0G. All microscopy experiments were analyzed in ImageJ either manually via kymograph analysis or automated by using the TrackMate plugin (*Tinevez et al., 2017*) unless otherwise described.

### Profilin binding affinity for actin monomers by fluorescence anisotropy competition experiments

To determine the equilibrium dissociation constant of profilin (wt or mutant proteins) and actin monomers from competition with another protein (the WCA domain of WAVE1) that binds to actin monomer with known affinity, the mean anisotropy values were plotted against the increasing total profilin concentration [nM]. Mean values were calculated from at least three measurements in three individual experiments per condition, error bars demonstrate the SD. The anisotropy data were fitted by an competitive binding model as described in *Wang (1995)* that analytically solves for the concentrations of the bound and free species from the known total concentrations of all proteins and the equilibrium dissociation constants for each of the two competing ligands:

(anisotropy as a function of the concentration of the profilin-actin complex):

$$r = r_f + (r_b - r_f)[PA] \tag{1}$$

The concentration of the profilin-actin complex can be determined from:
(concentration of the profilin:actin complex):

$$[PA] = \frac{[P]_0\left(2\sqrt{a^2 - 3b}\,\cos\left(\frac{\theta}{3}\right) - a\right)}{3K_P + \left(2\sqrt{a^2 - 3b}\,\cos\left(\frac{\theta}{3}\right) - a\right)} \tag{2}$$

and
(concentration of the WAVE1-WCA:actin complex):

$$[WA] = \frac{[W]_0\left(2\sqrt{a^2 - 3b}\,\cos\left(\frac{\theta}{3}\right) - a\right)}{3K_W + \left(2\sqrt{a^2 - 3b}\,\cos\left(\frac{\theta}{3}\right) - a\right)} \tag{3}$$

with

$$\theta = \cos^{-1}\frac{-2a^3 + 9a - 27c}{2\;\sqrt{(a^2 - 3b)^3}} \tag{4}$$

and

$$a = K_P + K_W + [P]_0 + [W]_0 - [A]_0 \tag{5}$$

and

$$b = K_P\left([W]_0 - [A]_0\right) + K_W\left([P]_0 - [A]_0\right) + K_P K_W \tag{6}$$

and

$$c = -K_P K_W [A]_0 \tag{7}$$

with $[A]_0$ being the total actin concentration, $[P]_0$ the total concentration of profilin, $[W]_0$ the total concentration of Atto488-WAVE1(WCA,) $K_P$ the equilibrium dissociation constant for the interaction between profilin and actin and $K_W$ the equilibrium dissociation constant for the interaction between Atto488-WAVE1(WCA,) and actin.

## Thymosin-β$_4$ binding affinity for actin monomers by fluorescence measurements

To determine the equilibrium dissociation constant of thymosin-β$_4$, the mean decrease in fluorescence intensity [au] was plotted against the increasing total thymosin-β$_4$ concentration. Mean values were calculated from at least three measurements in three individual experiments per condition, error bars demonstrate the SD. These data were fitted to a quadratic binding model as described in *Zalevsky et al. (2001)*:

$$I = I_f + \left(I_b - I_f\right)\frac{(K_D + [A] + [T]) - \sqrt{(K_D + [A] + [T])^2 - 4[A][T]}}{2[T]} \tag{8}$$

With $[A]$ being the total concentration of IEDANS-labeled actin, $[T]$ the total concentration of thymosin-β$_4$, $I_f$ and $I_b$ the fluorescent intensities in the free and bound state, respectively and $K_D$ being the equilibrium dissociation constant.

## Calculations of free species

To calculate the free actin, profilin, thymosin-β$_4$ (if added) and profilin-actin complex concentrations from the total concentration of actin, profilin and (if added) thymosin-β$_4$ in our TIRF-M single filament assays (see *Figure 2—figure supplement 1B*), we used an exact two species competition model as described in *Wang (1995)* and above (see Profilin binding affinity for actin monomers by fluorescence anisotropy competition experiments).

## Stopped flow measurements

For the determination of the association rate constant for profilin binding to actin monomers by tryptophan fluorescence quenching, the decrease in tryptophan fluorescence [au] was plotted against the total profilin concentration [μM]. The data were fitted with the following mono-exponential decay function to determine $k_{obs}$:

$$I(t) = \left(I_f - I_b\right) * e^{(-k_{obs}x)} + I_b \tag{9}$$

With I(t) the measured fluorescent intensity over time, $I_f$ and $I_b$ the tryptophan fluorescence in the free and bound state respectively and $k_{obs}$ being the observed reaction rate.

The association rate constants ($k_{on}$) were determined from linear regression fits of the $k_{obs}$ values as function of the total profilin concentration. The dissociation rate constants ($k_{off}$) were calculated from equilibrium dissociation constants ($K_D$) and association rate constants ($k_{on}$) using the following equation:

$$k_{off} = K_D * k_{on} \tag{10}$$

Errors for the dissociation rate constants were calculated using error propagation.

## Quantitative western blot analysis of total profilin and actin concentrations

Actin and profilin protein amounts per cell were quantified by western blot analysis using fluorescently-labeled secondary antibodies using a Odyssey Imaging System (LI-COR Biotechnology). The fluorescence signal intensity of the protein bands was analyzed from membrane scans using ImageJ. First, the detected intensity area was selected with the *rectangular tool*, for each protein intensity band (cellular protein and reference protein) an equal sized area was selected. Next, all lanes were plotted in an intensity plot profile reflecting the pixels across the selected area using the command *plot lanes*. The background signal intensity was subtracted from the protein intensity profile by drawing a straight baseline through the intensity curve representing the background intensity to the left and right of the curve. Then, the signal intensity (represented as the area under the intensity profile) was measured by selecting the *tracing tool* and clicking anywhere under the curve to integrate the intensity signal of the area of the plot profile. The measured intensities of the reference protein samples were plotted against the loaded protein mass [ng] and fitted with a linear function. The mass of the protein of interest was then calculated based on the slope of the reference protein. Finally, the protein concentration of actin/profilin was calculated as follows:

$$protein\ concentration = \frac{protein\ mass}{molecular\ weight * 0.5 * cell\ volume} \tag{11}$$

We assumed only half of the total cellular volume because actin and profilin are excluded from the endomembrane system (ER, Golgi, Mitochondria etc.) that occupies roughly 50% of the cell as measured by tomography methods. This means that in the most extreme case (all of the cell volume can be explored by profilin/actin), we are overestimating protein concentration by maximally 2-fold.

Because of the high affinity between ATP-bound monomers and profilin, we assumed that the cellular profilin-actin concentration must be close to the total profilin concentration. This is realistic as long as: i) the concentration of soluble ATP-bound actin is in excess over profilin so that profilin can be saturated with monomers, ii) the interaction between actin monomers and profilin is sufficiently rapid to approach thermodynamic equilibrium and iii) no other tight monomer binding proteins exist at high enough concentrations to effectively compete with profilin for actin monomer binding. While not all of these assumptions might strictly hold in the cellular environment, we believe that they still constitute reasonable approximations.

## Quantification of soluble profilin and actin concentrations

HT1080 wt and profilin/actin overexpressing cells were subjected to rapid pharmacological actin arrest as established in *Peng et al. (2011)*. Briefly, cells were treated with cell media containing 20 μM Y-27632 for 15 min at 37°C followed by the addition of 10 μM jasplakinolide and 15 μM latrunculin B (final concentration). Cells were treated independently with either jasplakinolide or latrunculin B only as controls.

After additional 20 min incubation, the cell medium was removed and the cells were lysed in lysis buffer (50 mM Tris pH 7.5, 150 mM NaCl, 1% Triton x-100, 1 mM MgCl2, 1 mM ATP, 1x protein inhibitor cocktail) for 5 min at 37°C. The cell lysate was initially centrifuged at 350xg for 5 min and the supernatant was separated into soluble and non-soluble/filamentous fractions by ultracentrifugation at 100.000xg and 15°C for 40 min. Next, the supernatant and pellet fractions were subjected to SDS-PAGE and further analyzed by western blot. Based on detected antibody signal intensity, fractions [% of total] of proteins in either the pellet or supernatant for both profilin and actin were quantified. Absolute soluble and filamentous protein concentrations were calculated from these fractional values and the total protein concentrations that were measured independently (see previous section). Errors are determined by error propagation.

## Cell volume measurements by fluorescence eXclusion

Data analysis was performed using custom written codes for MATLAB 2017b software written by QuantaCell. First, the raw GFP images were normalized following a manual cell tracking as it has been described earlier from *Cadart et al. (2017)*. For each cell type we analyzed $\geq$300 single cells. The cell volume distribution was plotted as a histogram and a lognormal distribution curve was fitted to the histogram. The mean volume [$\mu m^3$] and the error (SD) for each cell type was calculated.

## Barbed end elongation velocity from single filaments by TIRF-microscopy

Images were analyzed by manual filament tracking using the *segmented line tool* from ImageJ and further analyzed by the *kymograph plugin*. The slopes were measured to determine the polymerization rate of individual actin filaments. The pixel size/length was converted into microns/s. One actin monomer contributes to 2.7 nm of the actin filament length. For each experimental condition, the filament polymerization velocity was measured from $\geq$40 filaments from three independent experiments per condition and are reported as mean values with error bars representing SD. The elongation speed as a function of the total profilin-actin concentration were fitted by a hyperbolic model:

$$v([PA]) = \frac{v_{max}[PA]}{K_{0.5} + [PA]} \tag{12}$$

With [PA] being the total profilin-actin concentration, $v_{max}$ the maximal filament polymerization velocity at saturated profilin-actin concentrations and $K_{0.5}$ the profilin-actin concentration at half-maximal elongation speed.

## Velocity of single formin molecules in vivo

Data analysis was performed by manual filament tracking with the *segmented line tool* from ImageJ. Further, slopes from kymographs were measured to determine the moving rate of individual formins. The pixel size/length was converted into microns/s. One actin monomer contributes to 2.7 nm of the actin filament length. For each experimental condition $\geq$10 cells and $\geq$35 single molecules per cell were analyzed. Total number of molecules analyzed per condition was $\geq$650. All mean speed values were plotted as a histogram and fitted with a Gaussian function.

## Control experiments for single formin molecule in vivo imaging

HT1080 cells were seeded into 6-well LabTek dishes and transfected with a mNeongreen-tagged mDia2FH1-2 construct (see previous sections). 12–16 hr post-transfection, single formin molecules were either imaged in living cells or after cell lysis in 50 mM Tris pH 7.5, 150 mM NaCl, 1% Triton x-100 (~40.000 cells in 200 µl lysis buffer). Serial dilutions of cell lysate were added to a clean glass slide. All samples were imaged under the same imaging conditions (laser power, exposure time etc.) Intensity distributions and bleaching traces were analyzed using the TrackMate plugin from ImageJ.

## Acknowledgements

We thank Philippe Bastiaens for support and help in shaping the manuscript. Scott Hansen, Dyche Mullins, Thomas Surrey, Andrea Musacchio and Stefan Westermann for comments on the manuscript

and Marcus Taylor for EL-4 cells. This work was supported by HSFP (CDA00070/2017-2), the Max-SynBio network and the Max Planck Society.

## Additional information

### Funding

| Funder | Grant reference number | Author |
|---|---|---|
| Human Frontier Science Program | CDA00070/2017-2 | Johanna Funk<br>Peter Bieling |
| Max Planck Society | MaxSynBio | Johanna Funk<br>Peter Bieling |
| Bundesministerium für Bildung und Forschung | MaxSynBio | Johanna Funk<br>Peter Bieling |

The funders had no role in study design, data collection and interpretation, or the decision to submit the work for publication.

### Author contributions

Johanna Funk, Conceptualization, Data curation, Formal analysis, Methodology, Writing—original draft; Felipe Merino, Larisa Venkova, Pablo Vargas, Methodology, Writing—review and editing; Lina Heydenreich, Formal analysis; Jan Kierfeld, Formal analysis, Writing—original draft; Stefan Raunser, Matthieu Piel, Supervision, Methodology, Writing—review and editing; Peter Bieling, Conceptualization, Formal analysis, Supervision, Funding acquisition, Investigation, Writing—original draft, Writing—review and editing

### Author ORCIDs

Johanna Funk (iD) https://orcid.org/0000-0003-1214-8531
Felipe Merino (iD) http://orcid.org/0000-0003-4166-8747
Larisa Venkova (iD) http://orcid.org/0000-0001-5721-7962
Jan Kierfeld (iD) http://orcid.org/0000-0003-4291-0638
Stefan Raunser (iD) http://orcid.org/0000-0001-9373-3016
Peter Bieling (iD) https://orcid.org/0000-0002-7458-4358

### Decision letter and Author response

Decision letter https://doi.org/10.7554/eLife.50963.036
Author response https://doi.org/10.7554/eLife.50963.037

## Additional files

### Supplementary files

• Transparent reporting form DOI: https://doi.org/10.7554/eLife.50963.029

### Data availability

All data generated or analysed during this study are included in the manuscript and supporting files.

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

# Appendix 1

DOI: https://doi.org/10.7554/eLife.50963.030

## Kinetic model of actin filament growth in the presence of profilin

The simplest complete model for actin filament growth from actin monomers in the presence of profilin requires four reactions as shown in Appendix 1—Figure1.

Kinetic rates for reaction 3 (polymerization of actin in the absence of profilin) are well known from the literature (**Pollard, 1986**) and also obtained in this work; rates for reaction 4 (binding of profilin and monomeric actin in solution) are obtained in this work (see **Figure 3—figure supplement 1D**). Both sets of reaction rates are assumed to be known, and we use the values given in **Appendix 1—table 1** where needed. We aim to determine kinetic rates for the remaining reactions 1 and 2 for actin growth from actin-profilin and subsequent profilin release from fitting to growth velocity data (**Figure 2C**, **Figure 3E** in the main text). Some rates for reaction 1 (binding of profilin-actin complex to terminal protomer) have been obtained previously (**Courtemanche and Pollard, 2013**; **Pernier et al., 2016**) and are also given in **Appendix 1—table 1**.

**Appendix 1—table 1.** Model parameters.

| kinetic parameter | value | reference |
|---|---|---|
| $k_3$ | 11 $\mu M^{-1}s^{-1}$ | **Pollard, 1986** and this work |
| $k_{-3}$ | 1 $s^{-1}$ | **Pollard, 1986** |
| | 0.58 $s^{-1}$ | this work |
| $k_4$ | 40 $\mu M^{-1}s^{-1}$ | this work |
| $k_{-4}$ | 0.75 $s^{-1}$ | this work |
| $k_1$ | 11 $\mu M^{-1}s^{-1}$ | **Courtemanche and Pollard, 2013** |
| $k_{-1}$ | 50 $s^{-1}$ | **Courtemanche and Pollard, 2013** |
| | 5 $s^{-1}$ | **Pernier et al., 2016** |

DOI: https://doi.org/10.7554/eLife.50963.031

## Mathematical analysis

In the subsequent analysis we will assume that binding of profilin and monomeric actin in solution (reaction 4) has reached chemical equilibrium such that concentrations [P] of profilin, [A] of monomeric actin and [AP] of the actin-profilin complex are fixed and fulfill [A][P]/[AP] $=k_{-4}/k_4 = K_D$. Small measured values $K_D$ = 0.018 µM of the dissociation constant give rise to small values of [A] and [P] because of the high affinity of profilin for monomeric actin. For $[A]_{total} = [P]_{total}$, as employed in the experiments, we have [A] = [P] = $(K_D[AP])^{1/2} \approx 1.5$ µM for [AP]=100 µM. Shifts of this equilibrium by polymerization reactions 1 and 3 are negligible as long as the number of growing filament ends remains small. In the following we consider a single filament end.

## Growth velocity

The model allows an analytical solution for the mean growth velocity by considering the two complementary states P that profilin is bound to the filament and F that the filament end is profilin-free. The four reaction scheme induces a Markov process for transitions between these two states:

$$F \underset{k_{-2}+k_{-1}}{\overset{k_2[P]+k_1[AP]}{\rightleftharpoons}} P$$

In the stationary state, we find for the probabilities $p_P$ and $p_F$ to be in states P or F, respectively,

$$p_P = \frac{k_2[P]+k_1[AP]}{k_2[P]+k_1[AP]+k_{-2}+k_{-1}} = 1 - p_F \tag{A1.1}$$

The mean filament growth velocity in state P is $v_P = k_{-1} < 0$ because the only length changing process in the profilin-bound state is removal of an actin-profilin unit via reaction 1. The mean filament growth velocity in state F is $v_F = k_3[A] - k_{-3} + k_1[AP]$ because the length-changing processes are actin monomer addition and removal via reaction three and actin-profilin addition via reaction 1. The complete and exact result for the mean filament growth velocity v (in monomers per second) follows as

$$v = p_P v_P + p_F v_F = \frac{(k_3[A]-k_{-3}+k_1[AP])(k_{-2}+k_{-1})-k_{-1}(k_2[P]+k_1[AP])}{k_2[P]+k_1[AP]+k_{-2}+k_{-1}} \tag{A1.2}$$

by using (1). This is the main result of this appendix and in complete agreement with stochastic simulations of the four reaction model employing the Gillespie algorithm (**Gillespie, 1977**). If we assume equilibrium for actin-profilin binding in solution (reaction 4) and $[A] = [P] = (K_D[AP])^{1/2}$, and **Equation A1.2** yields

$$v = \frac{(k_1 k_{-2}[AP]) + (k_{-2}k_3 + k_{-1}k_3 - k_{-1}k_2)(K_D[AP])^{1/2} - (k_{-2}+k_{-1})k_{-3}}{k_1[AP]+k_2(K_D[AP])^{1/2}+k_{-2}+k_{-1}} \tag{A1.3}$$

## Michaelis-Menten kinetics in the limit of high profilin affinity for monomeric actin

Chemical equilibrium regarding actin-profilin binding results typically in [AP] >> [A], [P] because of the high affinity of profilin for monomeric actin (small $K_D$); this gives rise to several simplifications in the full result (2). Because [A] is small the polymerization pathway via actin monomer addition (reaction 3) is negligible, $k_3[A]$, k-3 << $k_1[AP]$ resulting in $v_F \approx k_1[AP]$. Moreover the rebinding of profilin to the filament end via reaction two is negligible because [P] is small, $k_2[P]$ << $k_1[AP]$, resulting in

$$v \approx \frac{k_1 k_{-2}[AP]}{k_1[AP]+k_{-2}+k_{-1}} = \frac{k_{-2}[AP]}{\frac{k_{-2}k_{-1}}{k_1}+[AP]} = p_P k_{-2} \tag{A1.4}$$

This is exactly analogous to Michaelis-Menten kinetics with

$$v \approx \frac{v_{max}[AP]}{K_M+[AP]} \tag{A1.5}$$

showing that, in the limit of high profilin affinity, the actin filament ends acts effectively as an enzyme for the cleavage of actin-profilin and justifying the use of hyperbolic Michaelis-Menten fits to describe growth velocity data in the presence of profilin throughout the paper. The result $v \approx p_P k_{-2}$ demonstrates clearly that profilin release is the rate-limiting step for polymerization and the profilin release rate $k_{-2}$ sets the maximal growth speed.

We can draw a number of conclusions from this result. In the limit of high profilin affinity only two parameters determine the growth velocity, the maximal growth velocity $v_{max} = k_{-2}$ and the Michaelis constant $K_M$. Therefore, we cannot expect to obtain more than two model parameters reliably from fitting growth velocity as a function of [AP]. In order to check whether the four reaction scheme satisfies detailed balance ($k_{-1}k_2k_3k_{-4} = k_1k_{-2}k_{-3}k_4$, see **Appendix 1—figure 1**), that is whether it requires external energy input, all rates have to be

known. Therefore, reliable statements about detailed balance are not possible from data on growth velocity as a function of [AP] only.

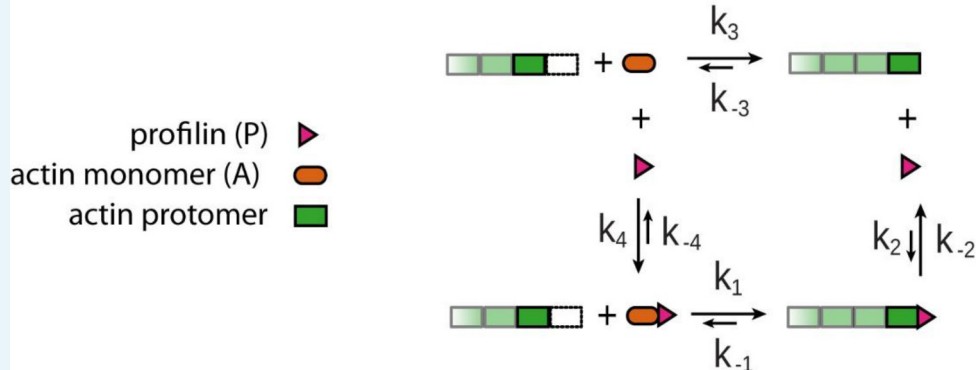

**Appendix 1—Figure 1.** Four reaction model for actin filament growth from actin monomers in the presence of profilin. 1)binding of profilin-actin complex to the terminal protomer, 2) profilin release from the terminal protomer, 3) binding of monomeric actin to the terminal protomer, 4) binding of profilin and monomeric actin to profilin-actin complex in solution. This four-reaction model completes the scheme from *Figure 1A* in the main text.
DOI: https://doi.org/10.7554/eLife.50963.032

For high actin-profilin concentrations, the growth velocity is given by $v \approx k_{-2} (1\, K_M/[AP])$ (i.e. a hyperbolic [AP]-dependence). The linear regime for small actin-profilin concentrations is given by $v \approx (k_{-2}/K_M)[AP]=k_{-2}k_1/(k_{-2}+k_{-1})$. This leads to the somewhat surprising conclusion that, also in the linear regime, the growth velocity depends on the profilin release rate $k_{-2}$. Therefore, profilin mutations will also change the linear increase of the actin filament growth velocity slightly at low profilin-actin concentrations (in agreement with *Figure 3E* in the main text).

## Fitting growth velocity data

From fitting growth velocity data with the Michaelis-Menten approximation (5) we can obtain the two parameters $v_{max} = k_{-2}$ and $K_M = (k_{-2} + k_{-1})/k_1$. Typically, we have $k_{-2} \gg k_{-1}$ such that $K_M \approx k_{-2}/k_1$, and we essentially obtain information on the two rates $k_{-2}$ and $k_1$ from these fits. *Appendix 1—table 2* shows results obtained from this procedure for all growth velocity data in *Figure 2C*, *Figure 3E* and *Figure 5C* in the main text.

**Appendix 1—table 2.** Results for rates $k_{-2}$, $k_1$ and $K_M$ from Michaelis-Menten fits of the growth velocity data in *Figure 2C*, *Figure 3E* (mutant profilin), and *Figure 5C* (with formins) in the main text.

|  | $k_{-2}$ [s$^{-1}$] | $K_M$ [µM] | $k_1$ [µM$^{-1}$s$^{-1}$] |
| --- | --- | --- | --- |
| β-actin-profilin | 558 ± 24 | 66 ± 3 | 8.4 ± 0.1 |
| β,γ-actin-profilin1 | 478 ± 31 | 54 ± 5 | 8.9 ± 0.3 |
| wt | 495 ± 15 | 57 ± 2 | 8.7 ± 0.1 |
| K125EE129K | 53 ± 7 | 26 ± 7 | 2.0 ± 0.3 |
| E82A | 609 ± 7 | 26.8 ± 0.7 | 22.7 ± 0.4 |
| R88K | 746 ± 21 | 35.7 ± 1.3 | 20.9 ± 0.3 |
| mdia1 | 1450 ± 167 | 22 ± 4 | 66 ± 5 |
| mdia2 | 820 ± 87 | 26 ± 4 | 31 ± 2 |
| daam1 | 573 ± 30 | 36 ± 3 | 16.1 ± 1.0 |

DOI: https://doi.org/10.7554/eLife.50963.033

We can also perform fits using the full result (3) (assuming equilibrium of profilin-actin binding in solution and [A] = [P] but not high affinity of profilin for monomeric actin) with four fit parameters $k_1$, $k_{-1}$, $k_2$, and $k_{-2}$ (and rates $k_3$, $k_{-3}$, $k_4$, and $k_{-4}$ taken from **Appendix 1—table 1**), We find that these fits are essentially insensitive to the additional parameters $k_{-1}$ and $k_2$, which is evidenced by a much higher variance for these parameters in least square fits. Therefore, these parameters cannot be reliably determined and the resulting fit does not improve over the Michaelis-Menten fit as demonstrated in **Figure 2** for one data set. Moreover, the insensitive rates $k_{-1}$ and $k_2$ can always be chosen such that detailed balance ($k_{-1}k_2k_3k_{-4} = k_1k_{-2}k_{-3}k_4$) is fulfilled by the whole reaction scheme without affecting the quality of the fit (see **Figure 2**) demonstrating that reliable statements about detailed balance are not possible.

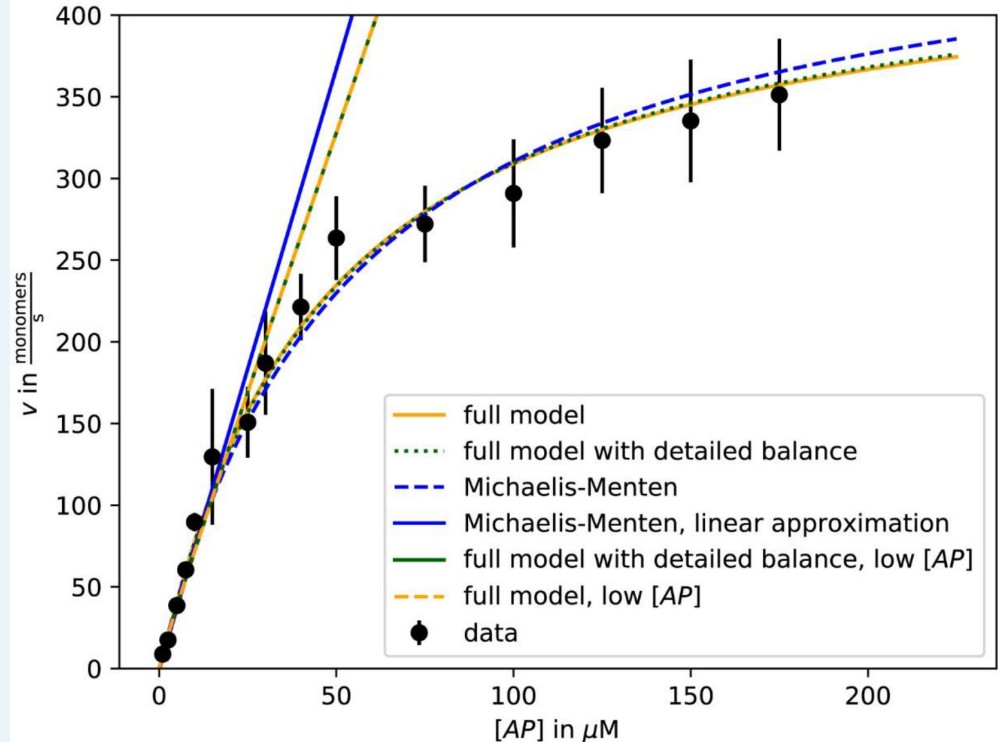

**Appendix 1—Figure 2.** Fit of growth velocity data as a function of [AP] for β-actin-profilin one from **Figure 2C** using either the Michaelis-Menten approximation (5) with $v_{max}$ and $K_M$ as fit parameters (dashed blue) and its linear low-[AP] approximation (solid blue), the full result (3) with $k_1$, $k_{-1}$, $k_2$, and $k_{-2}$ as fit parameters (solid yellow) and with the additional constraint of detailed balance on the fit parameters (dotted green) and the low-[AP] approximations to these results (dashed yellow, solid green).

DOI: https://doi.org/10.7554/eLife.50963.034

An important insight of the Michaelis-Menten fits of the growth velocity data is that the profilin mutations (**Figure 3E** in the main text) do not only alter the profilin release rate ($k_{-2}$). The fit results in **Appendix 1—table 2** reveal that also the binding rate of the actin-profilin complex to the terminal protomer $k_1$ changes. Because $k_1$ changes with mutation it is also not possible to fit all four growth velocity curves in **Figure 3E** with a common set of rates $k_1$ and $k_{-1}$ and variable profilin release rate $k_{-2}$ in order to determine $k_{-1}$.

