## [Decision Letter]

**Acceptance summary:**

Rapid actin filament assembly provides forces for cell migration, morphogenesis, and endocytosis. This study identifies a biochemical mechanism that kinetically limits actin filament elongation at physiological concentrations of assembly-competent actin monomers. The authors demonstrate that, in contrast to the 'textbook' model, the concentration of actin monomers is not the only parameter that controls the speed of actin filament growth. By studying actin filament assembly at physiological profilin-actin concentrations, they reveal that actin filament growth is limited by dissociation of profilin from the filament barbed end. Thus at high, physiological protein concentrations the speed of actin filament assembly becomes insensitive to the concentration of profilin-actin. Moreover, they show that formins stimulate filament growth by promoting profilin release from the filament barbed ends. Together, these and other data presented in the paper provide fundamental new insights into the molecular mechanisms that control actin filament elongation in cells.

**Decision letter after peer review:**

Thank you for submitting your article "Profilin and formin constitute a pacemaker system for robust actin filament growth" for consideration by *eLife*. Your article has been reviewed by three peer reviewers, including Pekka Lappalainen as the Reviewing Editor and Reviewer #1, and the evaluation has been overseen by Anna Akhmanova as the Senior Editor.

The reviewers have discussed the reviews with one another and the Reviewing Editor has drafted this decision to help you prepare a revised submission.

Summary:

Most eukaryotic cells exhibit a high cytoplasmic concentration of profilin-actin complexes. However, the mechanisms by which elongation of different actin filament structures are controlled under physiological conditions have remained elusive. Many studies have addressed this previously, but the work here by Funk et al. represents the first instance in which measurements were made at physiological profilin-actin concentrations.

They discover that – contrary to the long-held model – the rate of actin filament elongation saturates at high profilin-actin concentrations. Using profilin mutants with altered affinities for actin monomers and filament barbed ends, they provide strong evidence that at saturation the rate of actin filament elongation is limited by profilin release from the barbed end. By using a mutant version of actin, they provide evidence that profilin release form the filament barbed end does not require nucleotide hydrolysis of actin. They then show that formins accelerate elongation even at saturating concentrations of profilin-actin, at least in part by stimulating profilin release from filament barbed ends. Finally, they test the influence of profilin-actin concentrations in cells (by monitoring fluorescently labeled formins) to show that, as in vitro (at high concentrations), the rate of filament elongation is relatively insensitive to the concentration of profilin-actin, consistent with their model that profilin release limits elongation.

Overall, this is a very complete story and the conclusions are important for our understanding of actin filament growth in physiological (high profilin/actin) conditions. However, there are few relatively minor issues that should be addressed to further strengthen this interesting study.

Essential revisions:

The authors should more explicitly detail their assumptions about how the total concentration of profilin and actin is related to the concentration of soluble actin present as a profilin-actin species. This is especially important in their interpretation of the profilin-actin overexpression experiments. What evidence do the authors have that a 2-3 fold increase in profilin-actin expression results in a 2-3 fold increase in the concentration of soluble (profilin)-actin? Does overexpression influence the amount of filamentous actin in these cells?

Reviewer #2:

1) In Figure 2 C, the authors should include data points of actin filament elongation in presence of only actin monomers to show that indeed the growth velocity is linear in function of actin (only) at high concentration. The Figure 3E revealed that the assay does not limit the rate of elongation observed in Figure 1C, but it will be nice to include this control in Figure 1.

2) The Figures 3E/F and 5C are to me the most important figures. The authors instead of using a hyperbolic fit could use a (true) fit according to the kinetic model presented in Figure 7 top panel varying only k3 (in Figure 3E/F) and k3formin (in Figure 5C).

Alternatively, the authors could use hyperbolic fits in Figures 3E/F and 5C, but use a (true fit) in Figure 7 using their kinetic model.

3) The text and the legend of Figure 4 is probably misleading. ATP hydrolysis is very fast (as fast as actin polymerization), I believe that the authors want to demonstrate that the Pi release ADP-Pi == ADP (the limiting step in ATP==ADP) is not required for profilin release. This has been already demonstrated (maybe this figure could be moved to the supplement). However, the huge actin filament growth rate reported here (not limited by Pi release) suggests that unless capped by capping proteins or other proteins inhibiting elongation, growing actin filaments will be mostly loaded with ATP or ADP-Pi. The authors could better discuss this later point.

---

## [Author Response]

Essential revisions:The authors should more explicitly detail their assumptions about how the total concentration of profilin and actin is related to the concentration of soluble actin present as a profilin-actin species. This is especially important in their interpretation of the profilin-actin overexpression experiments. What evidence do the authors have that a 2-3 fold increase in profilin-actin expression results in a 2-3 fold increase in the concentration of soluble (profilin)-actin? Does overexpression influence the amount of filamentous actin in these cells?

Because of the high affinity between ATP-bound monomers and profilin, we assumed that the cellular profilin-actin concentration must be close to the total profilin concentration. This is realistic as long as: i) the concentration of soluble ATP-bound actin is in excess over profilin so that profilin can be saturated with monomers, ii) the interaction between actin monomers and profilin is sufficiently rapid to approach thermodynamic equilibrium and iii) no other tight monomer binding proteins exist at high enough concentrations to effectively compete with profilin for actin monomer binding. While not all of these assumptions might strictly hold in the cellular environment, we believe that they still constitute reasonable approximations given both our observations here and the general state of knowledge in the field.

The importance of this point, however, motivated us to perform additional experiments to address how much of profilin and actin in cells exists in a soluble form. Using a rapid pharmacological arrest of actin dynamics originally established by the Weiner lab (Peng et al., 2011) followed by fractionation of cell lysates through ultracentrifugation, we determined the relative fraction of soluble actin and profilin in wildtype and profilin-actin overexpressing HT1080 as requested.

In agreement with our assumptions, this analysis shows that all (>95%) of profilin exists in a soluble form, even in cells strongly overexpressing profilin-actin (new Figure 6—figure supplement 2C). Importantly, the soluble actin concentration increases by a very similar amount as the soluble profilin concentration in these overexpressing cells, demonstrating that an approximately 2-3-fold increase in the soluble profilin-actin concentration can indeed be assumed.

Interestingly, the amount of filamentous actin also rises in cells overexpressing profilin-actin. We assume that this is due to actin being more strongly overexpressed compared to profilin. We intentionally designed our multi-cistronic transgene with actin as the first coding sequence and profilin following two ribosomal skip sites. We believe that “excess” actin, which cannot be bound and soaked up by profilin will nucleate additional filaments resulting in the rise in the amount of filamentous actin we observe.

We included this new data in our manuscript (Figure 6—figure supplement 2C) and added additional text in the Results and Materials and methods part of the paper to explain which assumptions underlie our estimates of the soluble profilin-actin concentration.

Reviewer #2:1) In Figure 2C, the authors should include data points of actin filament elongation in presence of only actin monomers to show that indeed the growth velocity is linear in function of actin (only) at high concentration. The Figure 3E revealed that the assay does not limit the rate of elongation observed in Figure 1C, but it will be nice to include this control in Figure 1.

We agree with the reviewer that it would be desirable to include growth velocities for bare actin monomers at high concentrations to show that rates continue to scale linearly in the absence of profilin. Unfortunately, this is technically not feasible (even in our improved single filament assays) because of the rapid spontaneous nucleation of filaments at high concentrations (>20μM) of bare monomers. At these concentrations, monomeric actin converts to filaments within a few seconds or less! Even the most heroic efforts in literature (the classical 1984 Pollard paper comes to mind that measures the initial 10s after the initiation of growth) could not work with bare monomer concentrations >20μM for this very reason. This well-known kinetic instability of bare actin monomers is the main reason why cells can unlikely maintain them at levels significantly above the critical concentration. We have added text to the Results section of the paper to convey this point more clearly.

2) The Figures 3E/F and 5C are to me the most important figures. The authors instead of using a hyperbolic fit could use a (true) fit according to the kinetic model presented in Figure 7 top panel varying only k3 (in Figure 3E/F) and k3formin (in Figure 5C).Alternatively, the authors could use hyperbolic fits in Figures 3E/F and 5C, but use a (true fit) in Figure 7 using their kinetic model.

We agree with the reviewer that we should relate our data to the kinetic schemes we present in the paper. To improve this, we developed a comprehensive kinetic model of actin polymerization in the presence of profilin (see Appendix for detail). Under the conditions of our experiment, this full analytical model closely approximates the classical hyperbolic Michaelis Menten equation, which we therefore continued to use throughout the paper. Importantly, the full analytical model confirms that maximal rate of actin polymerization at saturated profilin-actin concentrations is identical to the dissociation rate of profilin from the barbed end.

Applying this model to our growth velocity data has, however, sparked some important additional insight specifically concerning the effect of the profilin mutants: We find that the mutations at the actin-profilin interface we use do not only change the dissociation rate of profilin, but also the *association rate* of the profilin-actin complex to the barbed end. This unexpected finding, however, does not alter our central conclusion that profilin release constitutes the kinetic limit at saturating profilin-actin levels.

We describe these findings both briefly in the main text and extensively in the new Appendix section.

3) The text and the legend of Figure 4 is probably misleading. ATP hydrolysis is very fast (as fast as actin polymerization), I believe that the authors want to demonstrate that the Pi release ADP-Pi == ADP (the limiting step in ATP==ADP) is not required for profilin release. This has been already demonstrated (maybe this figure could be moved to the supplement). However, the huge actin filament growth rate reported here (not limited by Pi release) suggests that unless capped by capping proteins or other proteins inhibiting elongation, growing actin filaments will be mostly loaded with ATP or ADP-Pi. The authors could better discuss this later point.

The reviewer’s comment made us realize that we should have done a better job explaining the results described in Figure 4. We agree with the reviewer that Pi release is not required for profilin release (since it is very slow) and that this has been convincingly demonstrated previously. However, the point here was to establish and use catalytically deficient actin mutant entirely unable to hydrolyze its associated nucleotide. The assays in Figure 4A-B show that this ATPase deficient actin cannot cleave the β-γ phosphodiester bond of its bound nucleotide with appreciable rates upon polymerization. This actin mutant therefore forms filaments that are homogenously and exclusively bound to ATP. ATPase deficient actin, however, can still polymerize with the same kinetics as wildtype actin even when bound to profilin, demonstrating that profilin release does neither require ATP hydrolysis nor Pi release.